

# Rapid recovery of Ediacaran oceans in the aftermath of the
# Marinoan glaciation
**Anthony Dosseto[1,*], Holly L. Taylor[1], Juraj Farkaš[2,3], Grant M. Cox[2], Andrew**
**Kingston[4], Andrew Lorrey[5], Alexander J. Corrick[2] and Bing Shen[6]**
*[1] Wollongong Isotope Geochronology Laboratory, School of Earth & Environmental*
*Sciences. University of Wollongong. Wollongong, NSW, Australia*
*[2] Department of Earth Sciences. University of Adelaide. Adelaide, SA, Australia*
*[3] Department of Environmental Geosciences, Czech University of Life Sciences, Prague,*
*Czech Republic*
*[4] National Institute of Water and Atmospheric Research. Wellington, New Zealand*
*[5] National Institute of Water and Atmospheric Research. Auckland, New Zealand*
*[6] Ministry of Education Key Laboratory of Orogenic Belts and Crustal Evolution, School of*
*Earth and Space Sciences, Peking University, Beijing 100871, People's Republic of China*
*\* corresponding author: tonyd@uow.edu.au*



**ABSTRACT**
The termination of Cryogenian glaciations would have undoubtedly impacted the
chemistry of Neoproterozoic oceans, with possible consequences for life; but the extent and
duration of this impact are poorly constrained. In this study, we use the lithium (Li) isotope
composition of Ediacaran cap dolostones from South Australia (Nuccaleena Formation) and
China (Doushantuo Fm) to investigate changes in ocean chemistry that followed the Marinoan
deglaciation. The effect of diagenesis was evaluated and while the Nuccaleena Fm is likely to
have preserved the primary composition of cap dolostone deposition, the offset in Li isotope
ratios observed for the Doushantuo Fm could possibly reflect partial overprinting by diagenetic
fluids. The Li isotope composition of Ediacaran seawater was estimated and we suggest it was
similar to that of late Cenozoic oceans for most of the cap dolostone deposition. Using a box
model for the oceanic Li cycle, we show that at the onset of deglaciation, the supply of riverine
Li to the oceans was up to 50 times the modern flux. The modelled riverine Li isotope
composition suggests that continents resembled modern high-latitude regions during this time.
This episode was short-lived (up to 1 Myr) and the subsequent supply of riverine Li was similar
to modern conditions, both in flux and isotope composition, for the whole duration of cap
dolostone deposition. These results suggest that Ediacaran oceans and continents rapidly
recovered from the Marinoan glaciation to reach environmental conditions similar to the late
Cenozoic. From the standpoint of the Li oceanic budget, the Ediacaran oceans in which
complex lifeforms emerged may have not been that different from our modern oceans.



## 1. Introduction


The Neoproterozoic era (1,000-542 Myr ago) is characterised by major environmental

changes including global glaciations (Hoffman et al., 2017), a second major increase in
atmospheric oxygen (Och and Shields-Zhou, 2012;Sahoo et al., 2016;Scott et al., 2008) and
the radiation of complex lifeforms (Knoll and Carroll, 1999). Biomarkers suggest that the first
metazoans emerged sometime in between the Sturtian and Marinoan glaciations (Love et al.,
2009), with algae replacing phototropic bacteria as the top marine primary producer between
659 and 650 Ma (Brocks et al., 2017). This shift has been suggested as resulting from a massive
supply of nutrients to the oceans at the end of the Sturtian glaciation (Brocks et al., 2017). The
end of the Marinoan glaciation could have triggered another major leap in evolution, since it is
followed by the appearance of centimetre-scale macroalgae and a significant increase in
metazoan complexity and diversity (Yuan et al., 2011;Yin et al., 2007;Yin et al., 2015).

The termination of the Marinoan glaciation is capped by the occurrence of dolostones,

termed "cap carbonates" (Rose and Maloof, 2010;Hoffman et al., 2017). These unique
formations are useful archives of changes in ocean chemistry that took place following the
Marinoan glaciation. For instance, their boron isotopic composition suggests ocean
acidification (Ohnemueller et al., 2014;Kasemann et al., 2005) while triple oxygen isotopes
show that atmospheric carbon dioxide levels were at their highest levels in the past 750 million
years (Bao et al., 2008). Calcium (Ca) isotopes show that the Ca supply to the oceans would
have been 14 to 140 times greater than the modern flux, and this was interpreted as a large
pulse in continental weathering (Kasemann et al., 2005). Zinc and cadmium isotopes hint at a
rapid resumption of primary productivity (Kunzmann et al., 2013;John et al., 2017). Huang et
al. (2016) have also used magnesium isotopes ($\delta^{26}Mg$) to suggest that a rapid pulse of
continental weathering took place prior to the end of the Marinoan glaciation, and subsided
following cap carbonate deposition. While their results provide some important insights on



environmental changes associated with large Neoproterozoic glaciations, because they focus
on siliclastic sediments the changes observed could reflect changes in the provenance of
sediments mobilised by erosion. For instance, the pulse in weathering inferred from high $\delta^{26}$Mg
values at the end of the Marinoan glaciation could reflect preferential erosion of sediments
from continental regions characterised by higher weathering intensity instead of a globally
more intense continental weathering. Furthermore, while Ca isotopes suggest a large element
flux from the continents, there is the need for testing this hypothesis with an unequivocal proxy
for silicate weathering. In order to fill this gap, we use the lithium (Li) isotopic composition of
Ediacaran cap carbonates to investigate possible changes in riverine inputs to the ocean that
reflect continental weathering.

Lithium isotopes ($^7$Li and $^6$Li) fractionate during the formation of secondary minerals,

with $^6$Li being preferentially incorporated into the solid, and thus have been used as a proxy
for silicate weathering (Burton and Vigier, 2011;Wimpenny et al., 2010;Pistiner and
Henderson, 2003;Verney-Carron et al., 2011;Decarreau et al., 2012). For instance, the Li
isotopic composition (noted $\delta^7$Li) of river waters records the extent of secondary mineral
formation at the catchment scale, where low $\delta^7$Li values are diagnostic of intense weathering
(Millot et al., 2010;Vigier et al., 2009). The Li isotopic composition of marine carbonates can
then be used to investigate how the riverine Li supply to the oceans has varied over time,
reflecting changes in continental weathering (Vigier and Goddéris, 2015;Li and West,
2014;Misra and Froelich, 2012;Wanner et al., 2014;Pogge von Strandmann et al., 2013;Lechler
et al., 2015;Pogge von Strandmann et al., 2017a). For instance, the increase in $\delta^7$Li values in
foraminifera throughout the Cenozoic has been interpreted as a consequence of the Himalayan
and Andean orogeneses, due to either a shift in continental weathering regime from transport-
limited, deeply weathered lowlands, to weathering-limited, poorly weathered steep terranes
(Misra and Froelich, 2012;Wanner et al., 2014;Vigier and Goddéris, 2015); or an increase in



submarine "reverse" weathering of authigenic clays (Li and West, 2014). For the former, it has
been proposed that the shift in continental weathering is represented either by a transition from
congruent to incongruent dissolution (Misra and Froelich, 2012), a decrease in weathering
intensity and increase in river suspended load (Wanner et al., 2014) or a decrease in Li storage
in secondary phases (Vigier and Goddéris, 2015). The $\delta^7$Li composition of calcite in chalk was
also used to evidence a large weathering pulse in the wake of the Ocean Anoxic Event 2
(OAE2), which could have contributed to a rapid recovery from the greenhouse state associated
with the OAE2 (Pogge von Strandmann et al., 2013). The $\delta^7$Li composition of Ordovician bulk
carbonates, brachiopods and shales showed that silicate weathering intensity was reduced
during the Hirnantian glaciation, resulting in inhibited $CO_2$ drawdown which could have
facilitated deglaciation (Pogge von Strandmann et al., 2017a).

In this study, Li isotope compositions were measured on two Marinoan cap carbonate

formations from Australia and China: the Nuccaleena Formation (Fm) and Member I of the
Doushantuo Fm, respectively. In combination with the recent determination of Li isotope
fractionation during Ca-Mg carbonate precipitation (Taylor et al., 2018), results are used to
model changes in the Li budget of Ediacaran oceans.

## 2. Study area

The Doushantuo Fm is part of the Yantgze block in China (Figure 1a), and was

deposited in a passive-margin setting (Jiang et al., 2011). We focused on the Lower Dolomite
Member (or Member I) which refers to ~5 m of massive and laminated dolomite beds at the
base of the Doushantuo Fm (Chen et al., 2004). Thirty-one samples were collected from core
14ZK at Daotuo village in Songtao county (28.131750 N, 108.892444 E WGS84), where the
cap dolostone is mainly composed of microcrystalline dolomite and dolomicrite with sheet-
crack structures (Huang et al., 2016).



The Nuccaleena Fm is located in South Australia and is part of the Adelaide Rift
Complex (Preiss, 1987), where Neoproterozoic sediments accumulated as a result of a
succession of rift and thermal subsidence phases (Jenkins, 1990). The Nuccaleena Fm was
sampled at Elatina Creek (31.357989 S, 138.617613 E WGS84; Figure 1b). It is underlain by
glacial diamectite and siltstone deposits of the Elatina Fm and is overlain by interbedded shales
and sandstones of the Brachina Fm. At Elatina Ck, we measured a thickness for the Nuccaleena
Fm of 4.2 m, in agreement with Raub (2008). Twenty-eight samples were collected in total:
three samples in the transition zone between the Elatina and Nuccaleena Fms (EC-1 to EC-3),
21 in the Nuccaleena Fm proper and four in the transition zone between the Nuccaleena and
Brachina Fms. The total thickness of section sampled was 6.75 m from the lowest to the highest
sample.
Font et al. (2010) suggested that the Nuccaleena Formation was deposited in only a few
100's of kyr; however, there are no radiometric dates that constrain the duration of deposition
of this formation. Condon et al. (2005) were able to bracket the duration of the Lower Dolomite
Member using ash beds, and it was estimated between 1.7 and 3.8 Myr. Numerical simulations
have suggested a duration for deglaciation of only 2 kyr (Hyde et al., 2000). Cap carbonate
deposition was proposed to have continued for several tens of thousands years afterward
(Creveling and Mitrovica, 2014), although this is significantly shorter than the proposed
duration by Condon et al. (2005). Both sections were correlated assuming synchronicity of
deposition and considering a total thickness of 5 and 4.2 m for the Lower Dolomite Member in
core 14ZK and the Nuccaleena Fm at Elatina Ck (Raub et al., 2007), respectively.

**3. Methods**
*3.1. Mineralogy*



Mineralogical compositions were determined on bulk samples of the Nuccaleen Fm at
University of Wollongong. This was undertaken by powder X-ray diffraction (PXRD) of finely
ground aliquots performed on a PANalytical X'Pert PRO diffractometer outfitted with a Co-
target tube (operated at 40 kV and 40 mA), a high-speed Scientific X´Celerator detector, 0.5°
antiscattering and divergence slits, spinner stage, primary and secondary soller, as well as an
automatic sample changer. Samples were finely ground by hand using a mortar and pestle prior
to analysis and were loaded in a random orientation using the top loading technique. The
samples were analyzed over the range $4 - 85°$ $2\theta$ with a step size of $0.008°$ $2\theta$ and a count time
of 40 seconds/step. Mineral quantification was obtained by Rietveld Refinement of the XRD
patterns using the PANalytical X'Pert HighScore Plus Software and its implemented pdf-2
database.

*3.2. Oxygen and carbon isotopic compositions*
Ground bulk samples were prepared and analysed for oxygen and carbon isotopes at
the National Institute of Water and Atmospheric Research. Each sample was reacted with 3
drops of $H_3PO_4$ at 75°C in a Kiel automated individual carbonate reaction device coupled with
a mass spectrometer. Either a Kiel III coupled to MAT 252 mass spectrometer or a Kiel IV
coupled with MAT 253 mass spectrometer were used to derive $^{18}O/^{16}O$ and $^{13}C/^{12}C$
measurements. All values are reported relative to Vienna Pee Dee Belemnite (VPDB), where
$\delta^{13}C$ has a value of +1.95‰ and $\delta^{18}O$ has a value of -2.20 ‰ for NBS19 calcite. Internal
precision of measurements is 0.02-0.08 ‰ for $\delta^{18}O$ and 0.01-0.06 ‰ for $\delta^{13}C$, external
precision is 0.03‰ for $\delta^{18}O$ and 0.02‰ for $\delta^{13}C$, relative to VPDB.



*3.3. Lithium isotopic compositions and element concentrations*

Sample preparation for Li isotope and trace element concentration measurement was

undertaken in a Class 100 cleanroom at the Wollongong Isotope Geochronology Laboratory,
University of Wollongong. Because dolostones contain a significant proportion of siliclastic
material, samples were leached following the protocol recently developed by Taylor et al.
(2018): one gram of ground rock was weighed and leached with 0.05M HCl at room
temperature for 1 hr. Samples were then centrifuged and the supernatant was weighed, dried
down and re-dissolved in a weighed amount of 0.3M $HNO_3$. Aliquots of solution were taken,
weighed and diluted 100 times for Ti, Mn, Rb, Sr and Li concentration determinations and
10,000 times for Mg, Ca and Al concentrations. The remainder was dried down, and re-
dissolved in 1M HCl. A volume of solution equivalent to ~60 ng of Li was processed for cation
exchange chromatography procedure, as described in Balter and Vigier (2014); calibration of
each column was assessed using seawater (Taylor et al., 2018).

Element concentrations were determined by single collector ICP-MS (Inductively

Coupled Plasma Mass Spectrometry) on a ThermoFisher iCAP-Q at the Wollongong Isotope
Geochronology Laboratory, University of Wollongong. Samples were introduced in 0.3M
$HNO_3$ using a concentric nebuliser and cyclonic spray chamber. Standard nickel cones were
used. Concentrations were calculated using an external calibration curve, established by
analysing seven calibration standards (Inorganic Ventures 71A) with concentrations ranging
from 0.05 to 100 ppb. Instrument blank correction was applied during analysis. Drift correction
was performed using element standard 71D from Inorganic Ventures.

Lithium isotope ratios were measured by MC ICP-MS (Multi Collector Inductively

Coupled Plasma Mass Spectrometry) on a ThermoFisher Neptune Plus at the Wollongong
Isotope Geochronology Laboratory, University of Wollongong. A 30 ppb solution of IRMM-
16 Li isotopic standard was used at the start of each session to tune the instrument. An intensity

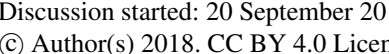



of ~1 V was routinely obtained for $^7$Li, while the background $^7$Li intensity was between 5-30
mV. During analysis, standard bracketing, using IRMM-16 as the primary standard, was
applied to correct the measured $^7$Li/$^6$Li for mass bias (Flesch et al., 1973). The accuracy of
analysis was assessed using synthetic solutions Li6-N and Li7-N as secondary standards
(Carignan et al., 2007) every 6 samples. Before each standard and sample, the instrument blank
was measured and subtracted from each isotope. The $^7$Li/$^6$Li ratios are expressed as $\delta^7$Li values
using L-SVEC to normalize the isotopic ratio (Carignan et al., 2007).

**4. Results**

For the Nuccaleena Fm, most samples show dolomite contents between 70 and 83 wt.

% (Figure 2). However, four samples (two at the bottom, two at the top of the section) show
values as low as 33 wt. % (Table 1). These samples are also characterised by higher
concentrations of quartz (up to 32 wt. %) or muscovite (up to 21 wt. %) (Figure 2). They are
derived from the Elatina-Nuccaleena and Nuccaleena-Brachina transitions; thus these
mineralogical compositions likely reflect waning detrital input to the Ediacaran ocean at the
onset of cap carbonate formation, and an increasing siliclastic contribution at its end, eventually
dominated by siltstones of the Brachina Fm. Other minerals present are calcite (≤17 wt. %),
ankerite and albite (both ≤11 wt. %), chlorite (≤6 wt. %) and kaolinite (≤3 wt. %) (Table 1).
Oxygen ($\delta^{18}$O) and carbon ($\delta^{13}$C) isotope compositions range from -8.90 to -6.29 ‰ and from
-3.19 to -1.06 ‰, respectively (Table 2). These values are similar to previously published data
on the Nuccaleena Fm for the same section (Rose and Maloof, 2010) and at two other locations
(Kunzmann et al., 2013;Hoffman and Schrag, 2002).

For the Doushantuo Fm, $\delta^{18}$O and $\delta^{13}$C values range from -12.15 to -6.93 ‰ and -6.24

to -3.25 ‰, respectively (Table 3). A similar range was previously reported by Jiang et al.
(2003) and Zhou et al. (2004) at other locations (excluding the exotic C isotopic compositions




of clotted microcrystalline limestones and peloids in Jiang et al. (2003)). Both Nuccaleena and
Doushantuo Fms show upward decreasing $\delta^{13}C$ values (Figure 3), in agreement with previous
observations on Marinoan cap carbonates (e.g. Zhou et al. (2004)). This has been  proposed to
reflect a massive alkalinity flux to the oceans resulting from rapid post-glaciation weathering
(Hoffman et al., 1998). This hypothesis has been challenged by a lack of significant variation
in $^{87}Sr/^{86}Sr$ throughout the cap carbonate sequence (Kennedy et al., 2001), although recent
studies have shown that a post-glacial meltwater is consistent with Sr isotopes or rare earth
element compositions (Verdel et al., 2018;Liu et al., 2014). Oxygen and C  isotope
compositions correlate positively (Figure 4), also consistent with previous studies and
interpreted as diagenetic overprint of primary compositions (e.g. Jiang et al. (2003)).

Lithium isotopic composition (expressed as $\delta^7Li$ ) in leaching solutions range from 2.9

to 13.7 ‰ in the Nuccaleena Fm (Table 2) and from 3.0 to 13.0 ‰ in the Doushantuo Fm
(Table 3). This range is at the lower end of values encountered in Phanerozoic carbonates
(Pogge von Strandmann et al., 2017a;Pogge von Strandmann et al., 2013;Lechler et al.,
2015;Sun et al., 2018;Misra and Froelich, 2012). While ranges of values are similar in both
sections, the Doushantuo Fm shows values systematically lower than those of the Nuccaleena
(Figure 5).

**5. Discussion**
*5.1. Dolomitisation and diagenesis*

Several mechanisms have been proposed for cap dolostone formation: early, late, deep

burial dolomitisation, or direct precipitation (Fairchild and Kennedy, 2007). Based on Mg and
Sr isotope records in the Nuccaleena Fm also sampled at Elatina Creek, and in Mongolian cap
dolostones, Liu et al. (2014) have argued that direct precipitation or early dolomitisation are
the most likely scenarios. The Li isotope composition of dolostones could thus reflect that of



the seawater at the time of formation. This is supported by Kunzmann et al. (2013) who
suggested that the Nuccaleena Fm faithfully preserves post-glacial seawater composition based
on carbon and oxygen isotopic compositions. They indicated that while most dolomitized
carbonates are commonly coarsely recrystallised and preferentially occur at the top of shoaling
sequences, the Nuccaleena and other equivalent cap dolostones are characteristically fine-
grained and were deposited at the base of a transgressive-regressive sequence (Hoffman et al.,
2007). Rose and Maloof (2010) also showed consistent carbon isotopic compositions at the
1km to 100km scale in the Adelaide Rift Complex, arguing against a diagenetic overprint.
Conversely, Jiang et al. (2003) suggested that the positive relationship between O and C isotope
compositions in the Doushantuo Fm could reflect a diagenetic overprint of primary
compositions. Huang et al. (2016) have suggested that the cap dolostone from core 14ZKZ has
experienced little diagenetic alteration since it is composed of dolomicrite and microcrystalline
dolostone without significant recrystallization.

To further assess the impact of diagenesis on the sections sampled, we measured

oxygen and carbon isotopic compositions on bulk rock samples of both Nuccaleena and
Doushantuo Fms. Measured compositions show a positive relationship between $\delta^{18}O$ and $\delta^{13}C$
for the Doushantuo Fm but not for the Nuccaleena (Figure 4), suggesting that while the former
might have been affected by diagenesis, the latter may have not, in agreement with previous
studies. Lighter $\delta^{7}Li$ values in Doushantuo cap dolostones compared to the Nuccaleena Fm
(Figure 5), could thus be the result of a partial diagenetic overprint, or local Li isotope signal
due to variable proportions of seawater-freshwater mixing between these two sites.

The effect of diagenesis on Li isotopes was also tested by comparing $\delta^{7}Li$ values to

Mn/Sr ratios in the leaching solutions. The Mn/Sr ratio is often used an index for diagenetic
alteration (Brand and Veizer, 1980), since diagenesis can result in a carbonate uptake of Mn.
There is no clear relationship between $\delta^{7}Li$ compositions and Mn/Sr ratios for either formation



(Figure 6), suggesting that while diagenesis may have imparted the observed range in Mn/Sr
values (in particular for the Nuccaleena Fm, which shows relatively high values), it did not
have a measureable nor systematic effect on Li isotope compositions. Liu et al. (2013) observed
a negative relationship between $^{87}Sr/^{86}Sr$ ratios and $\delta^{18}O$ values for Nuccaleena cap dolostones,
arguing that diagenetic fluids introduced radiogenic $^{87}Sr$ and light oxygen isotopes. Here, there
is no relationship between $\delta^7Li$ and $\delta^{18}O$ in either section (Figure 7), suggesting that if
diagenesis resulted in an enrichment in $^{16}O$, there was no associated systematic modification
of the Li isotope ratio.

*5.2. Lithium isotopes in Ediacaran cap carbonates and the evolution of the Neoproterozoic*

*oceans*

In the Nuccaleena Fm, $\delta^7Li$ values range from 7.3 to 13.7 ‰ (excluding transition

zones), which contrasts with the lower values encountered in the transition zone between the
Elatina and Nuccaleena Fms (<5 ‰). While Li isotope compositions were measured in leaching
solutions and should be devoid of silicic-bound Li (Taylor et al., 2018), it is possible that low
$\delta^7Li$ values in solutions of the Elatina-Nuccaleena transition zone reflect the contribution of
isotopically light Li from the silicic fraction, more abundance in this part of the section (Figure
2). Interestingly, while the Nuccaleena-Brachina transition zone is also characterised by a
higher silicic content (Figure 2), measured $\delta^7Li$ compositions in leaching solutions (6.2-11.7
‰) are similar to those of the Nuccaleena Fm. Consequently, it is possible that the low $\delta^7Li$
values in the Elatina-Nuccaleena transition zone faithfully reflect the composition of the
carbonate fraction, and thus of the ocean at the time. In Lower Dolomite Member of the
Doushantuo Fm, $\delta^7Li$ compositions are overall lower than those in the Nuccaleena Fm by ~2
‰ (calculated by comparing averages for each formation, excluding $\delta^7Li$ values in transition



zones for the Nuccaleena Fm). As indicated above, this offset could possibly be the result of
diagenetic alteration of the Doushantuo Fm.

Lithium isotope compositions of the cap dolostones show little variations throughout

each section (excluding transition zones), probably indicative the Li oceanic budget did not
experience any major changes during cap dolostone deposition. This is discussed further below.
Values are at the lower end of the range of $\delta^7$Li values observed for other carbonates (Pogge
von Strandmann et al., 2017a;Pogge von Strandmann et al., 2013;Lechler et al., 2015;Pogge
von Strandmann et al., 2017b;Misra and Froelich, 2012). However, such comparison has little
value since previous studies have focused on calcium carbonates, and the extent of isotopic
fractionation between solution and carbonates is different whether considering calcium or Ca-
Mg carbonates (Marriott et al., 2004a;Marriott et al., 2004b;Taylor et al., 2018). The abundance
of dolomite in the geological records, compared to its paucity in modern environments and the
difficulty to produce it experimentally at low temperature, led to coining the term 'dolomite
problem' (e.g. McKenzie, 1991;Purser et al., 1994). Microbial mediation has been proposed as
a key mechanism for dolomite formation in natural environments (Vasconcelos et al.,
1995;Wright, 1999;Wacey et al., 2007;Burns et al., 2000), although Arvidson and Mackenzie
(1999) have shown that changes in seawater temperature and chemistry could account for the
abundance of dolomite in the geological record. While the precipitation experiments in Taylor
et al. (2018) were performed in absence of microbial mediation, they offer an estimate of the
Li isotopic fractionation between solution and Ca-Mg carbonate, $\alpha_{prec-sol}$:
$$10^3 ln\alpha_{prec-sol} = -\frac{(2.56 \pm 0.27)\times10^6}{T^2} + (5.8 \pm 1.3)$$     (1)
where $T$ is the temperature of precipitation (in K).

The relationship above allows us to use the Li isotope composition of cap dolostones

to estimate the Li isotope composition of Ediacaran seawater (and/or of the seawater-
freshwater mixture), assuming the temperature of seawater during cap dolostone formation.



Knauth (2005) suggested that seawater temperature dropped by 10-15 °C between 685 and 550
Ma to reach Phanerozoic values (10-20 °C). Considering a seawater temperature of 10 or 40
°C would yield isotopic fractionation factors within error of each other ($\alpha$ = 0.974 ± 0.241 or
0.980 ± 0.243, respectively). The $\delta^7$Li composition of seawater was calculated for temperatures
of cap dolostone formation of 10, 25 and 40 °C (Tables 2 & 3). In the Nuccaleena Fm, average
calculated seawater $\delta^7$Li values range from 30.3 (with a seawater temperature at 40 °C) to 36.2
‰ (at 10 °C) (Figure 8). These values are similar to the modern seawater composition (31.1 ±
0.2 ‰; (Jeffcoate et al., 2004)). This observation suggests that following deglaciation, the
chemistry of Ediacaran oceans evolved rapidly towards that of modern oceans (at least from
the standpoint of the Li cycle), since even the base of the cap dolostones suggest modern-like
seawater $\delta^7$Li compositions. The Elatina-Nuccaleena transition zone shows isotopic
compositions that are much lighter (average values ranging from 23.9 to 29.8 ‰, depending
on the seawater temperature considered). As discussed above, this could indeed reflect
isotopically lighter compositions for the seawater at that time, illustrating the recovery in the
aftermath of the Marinoan glaciation, or result from an incomplete isolation of the carbonate-
bound Li during sample preparation.

Calculated seawater $\delta^7$Li compositions are investigated further using a box model for

the oceanic Li cycle similar to that described in previous studies (Pogge von Strandmann et al.,
2013;Lechler et al., 2015). The seawater Li budget is written as follows:
$\dfrac{dN_{Li}}{dt} = F_r + F_h - F_{sed}$                                                    (2)
where $N_{Li}$ represents the mass of Li in seawater (in Gmol), $F_r$ and $F_h$ are the input fluxes of
riverine and hydrothermal Li, respectively (in Gmol/yr). $F_{sed}$ is the output flux of Li, lumping
together uptake into marine sediments and alteration of oceanic crust. $F_{sed}$ is taken to scale to
$N_{Li}$ via a constant partition coefficient $k$ (Lechler et al., 2015): $F_{sed} = k$ x $N_{Li}$.

The evolution of the seawater Li isotope ratio is written as follows:





$N_{Li} \frac{dR_{SW}}{dt} = F_r(R_r - R_{SW}) + F_h(R_h - R_{SW}) - F_{sed}(R_{sed} - R_{SW})$        (3)
where $R_{SW}$, $R_r$ and $R_h$ are the $\delta^7$Li compositions of seawater, the riverine and hydrothermal
inputs, respectively. $R_{sed}$ is derived from $R_{SW}$ and assuming a constant isotopic fractionation
between marine sediments and seawater of 16 ‰ (Huh et al., 1998;Misra and Froelich, 2012).

Our model does not consider an output flux of Li via subduction, similarly to Pogge

von Strandmann et al. (2013);Lechler et al. (2015). The model starts at the onset of the
Marinoan glaciation and the ocean is assumed to have an initial mass of Li similar to that of
the modern ocean: $3.6 \times 10^7$ Gmol (Misra and Froelich, 2012), for lack of a better constraint. In
the same way, we assume modern values for the hydrothermal flux of Li and its isotopic
composition: 13 Gmol/yr and 8.3 ‰, respectively (Misra and Froelich, 2012). These values are
kept constant throughout the Marinoan glaciation and Ediacaran cap carbonate deposition. If
the hydrothermal flux was greater than the modern value (Gernon et al., 2016), riverine Li
fluxes would be required to be greater than the ones calculated below. The duration of the
Marinoan glaciation is taken to be 15 Myr (Hoffman et al., 2017). The seawater $\delta^7$Li
compositions used in the model are those calculated for a temperature of 25 °C for cap
dolostone formation, since we show above that the temperature does not affect greatly
calculated seawater compositions. The duration of cap dolostone deposition is taken to be 3
Myr (Condon et al., 2005). A much shorter duration has been proposed (e.g. Creveling and
Mitrovica, 2014;Higgins and Schrag, 2003) and model results for various durations are
discussed below. However, a duration of 3 Myr is considered in most scenarios below as it
leads to the most conservative estimates of riverine Li fluxes and isotopic compositions.

We first tested the effect on the model's results of a delay between the end of the

Marinoan glaciation and the onset of cap carbonate deposition, as Huang et al. (2016) have
suggested there is a pulse of continental weathering ~1 Myr prior to the onset of deposition of
the Doushantuo Fm. We modeled two cases: one where the hydrological cycle resumed at the





onset of cap carbonate formation (scenario 1; Table 4 & Figure 9a), and another one where it
preceded cap carbonate formation by ~1 Myr (scenario 2; Table 4 & Figure 9b). In the first
case, a short pulse (0.1 Myr) of intense weathering is required, where the riverine Li flux would
have been 50 times the modern value, and the riverine $\delta^7$Li would be 35 ‰. This large riverine
flux is dictated by the need to increase from seawater $\delta^7$Li values as low as ~25 ‰ at the end
of the Marinoan glaciation, to 30-35 ‰ at the base of the cap dolostone. Following this stage,
the riverine Li flux would still need to be 5 times the modern value, with a riverine $\delta^7$Li similar
to the modern average river, for the remainder of cap dolostone deposition. In the second case,
the million year-long stage preceding cap dolostone deposition would be characterised by a
riverine Li flux three times the modern value and a high riverine $\delta^7$Li (30 ‰). During cap
dolostone deposition, the riverine Li flux and isotope composition could have been similar to
modern values.

In most scenarios, we assumed that the riverine input of Li shut down during glaciation.

This results in a marine Li budget at the end of the glaciation about 1/3 of that at the start. If
we consider there was a delay between the end of deglaciation and the onset of cap dolostone
deposition, this assumption has little impact on the model results. Even if we assumed that the
riverine Li flux during glaciation was as high as 80% of the modern value, resulting in a much
higher seawater $\delta^7$Li at the end of the glaciation (~30 ‰), the model can fit the data with the
same parameters (scenario 3; Table 4 & Figure 10b) as when considering a complete
hydrological shutdown (scenario 2). However, if considering no delay between the end of
glaciation and the onset of cap dolostone deposition, a riverine Li flux during glaciation 80%
that of the modern value only requires the riverine Li flux to be 10 times the modern value
during the pulse of riverine Li supply at the onset of cap dolostone deposition (scenario 4;
Table 4 & Figure 10a).



As mentioned above, several authors have suggested that cap carbonate deposition
occurred over a period of a few 100,000 years or less (e.g. Creveling and Mitrovica,
2014;Higgins and Schrag, 2003). If we consider a duration of 300,000 yr for cap carbonate
deposition (scenario 5; Table 4 & Figure 12), the model parameters are very similar to those
obtained with a 3 Myr duration (scenario 2). The only notable differences are that the riverine
Li flux during cap carbonate deposition would be twice the modern value, and its isotopic
composition would high (35 ‰). Thus, uncertainty on the duration of cap carbonate deposition
does not significantly affect the model results.
The model above assumes a well-mixed ocean throughout glaciation, deglaciation and
cap carbonate deposition. An alternative scenario has been invoked to explain strontium and
magnesium isotope compositions of the Nuccaleena Fm where the Ediacaran ocean would have
experienced stratification as a consequence of glacial meltwater pulses (Liu et al., 2013;Liu et
al., 2014;Shields, 2005). We have considered such a scenario: in this case, the ocean is well
mixed during stage 1 (Marinoan glaciation); however, during stage 2 (time lag between
deglaciation and onset of cap carbonate deposition), the shallow ocean only receives riverine
Li (meltwater plume), while the deep ocean only receives hydrothermal Li and is the only
compartment that loses Li to authigenic sediments and oceanic crust alteration. During stage 3
(cap carbonate deposition), shallow and deep oceans are assumed to mixed instantaneously and
then experience input and loss of Li as a well-mixed ocean for the remainder of the stage
duration. More complex scenarios of stratification and overturn could be considered, however
there is no data justifying such hypothesis. In the model, we consider that when the ocean is
stratified, the shallow ocean represents 20% of the total oceanic volume (scenario 6; Table 4
& Figure 12). Values ranging between 10 and 50% yield identical results (not shown). This
lack of sensitivity to the size of the shallow and deep compartments mainly stems from the
short duration of isolation. Model results are very similar to the well-mixed ocean scenario



(scenario 2), the only notable difference being that the stratified ocean scenario requires a
riverine Li flux during cap carbonate deposition five times greater than the modern value.

Overall, the model suggests a pulse of riverine Li that lasted 0.1 to 1 Myr, and was three

to 50 times greater than the modern flux. In all scenarios, the Li isotopic composition of the
average riverine input during this stage was ~30-35 ‰. This value is comparable to the
composition of modern Icelandic rivers (Pogge von Strandmann et al., 2006;Vigier et al.,
2009). During cap carbonate deposition, most scenarios suggest that the pulse in riverine Li
was followed by a flux comparable to the modern value, and with an isotopic composition also
comparable to that of the average modern river, yielding Li isotope compositions for the
Ediacaran ocean similar to those observed since the Pliocene.

In the aftermath of the Marinoan glaciation, new rivers would thus have drained

landscapes similar to modern high latitude regions. This type of environment would have
dominated exposed continents for up to 1 Myr. After that, Ediacaran landmasses could have
already experienced climatic and environmental conditions similar to those experienced in the
late Cenozoic. This model suggests that Ediacaran oceans and continents rapidly recovered
from the deglaciation, and within a million year of the end of the Marinoan glaciation, Earth's
surface was possibly resembling our modern environment. The large flux of riverine Li during
deglaciation would have been accompanied by a large supply of other nutrients. If this nutrient
pulse lasted one million years (scenarios 2, 3, 5 & 6), it could have been the key for the
development of complex lifeforms since substantial "bursts" in evolution require million-year
long changes (Uyeda et al., 2011).

**6. Conclusions**

The lithium isotope composition of cap dolostones of the Nuccaleena (Australia) and

Doushantuo (China) Fms provide insights into the environmental changes which occurred in



the aftermath of the Marinoan glaciation. Several lines of evidence argue for a primary origin
for the dolomite of these cap carbonates, which are thus believed to inform on the chemistry of
the Ediacaran oceans. The Nuccaleena Fm shows no significant signs of diagenetic alteration
and its Li isotope composition is likely to reflect isotopic fractionation during cap carbonate
deposition. It is less clear for the Doushantuo Fm, and the ~2 ‰ offset in $\delta^7$Li values compared
to the Nuccaleena Fm could possibly be the result of partial overprinting by diagenetic fluids.

The estimated Li isotope composition of Ediacaran seawater was used in combination

with an ocean box model to investigate changes in the oceanic Li cycle. Most scenarios suggest
a pulse of riverine Li flux up to 50 times the modern value, during the deglaciation, with a
riverine $\delta^7$Li similar to that of rivers in modern high latitude regions. This episode lasted up to
1 Myr, after which the supply of riverine Li would have been similar to what oceans have
experienced since the Pliocene. These results suggest that Ediacaran oceans and continents
rapidly recovered from the Marinoan glaciation, and within 1 Myr of the onset of deglaciation
the planet returned to a greenhouse state. The million-year pulse of continent-derived nutrients
could have promoted the emergence of complex lifeforms, which would have thrived in an
Earth surface system possibly not that different from that of the late Cenozoic.

**Author contribution**
AD and JF designed the project. AD, HLT, JF, AJC conducted fieldwork in Australia, BS
provided Chinese samples. HLT conducted the element concentration and lithium isotope
analyses. AK conducted oxygen and carbon isotope analyses. AD, JF, GMC and BS interpreted
the results. AD conducted the numerical modelling and wrote the manuscript. All authors
edited the manuscript.

**Acknowledgments**



We would like to thank Lili Yu and Leo Rothacker for help in collecting the Li isotopic
data, and Magali Roux for help in the field and comments on the manuscript. HT acknowledges
an Australian Postgraduate Award. We also thank Arthur Coulthard, the Ikara-Flinders Ranges
National Park and the Department of Environment, Water and Natural Resources (DEWNR)
for granting us permission to collect samples from Elatina Creek (research permit Y26506-2).
Fieldwork for this work was funded by a GeoQuEST grant to AD. Stable isotope analyses were
supported by the NIWA core funded project "Climate Present and Past".





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



**Figure captions**
Figure 1. (a) Map showing the Doushantuo Fm in the context of a simplified geological map
of southern China. The blow-up inset shows the location of the studied core (14ZK). Modified
from Huang et al. (2016). (b) Simplified geological map of the central Flinders Ranges,
Australia (inset: the square shows the location of the Flinders Ranges in Australia). The
sampling site (Elatina Creek) is indicated as a red circle. Modified from Maloof et al. (2010).
Figure 2. Dolomite (green circles), muscovite (red) and quartz (blue) concentrations in cap
carbonates of the Nuccaleena Formation, as a function of the height above the base of the
formation. The curves are polynomial fits shown with 95% confidence intervals (grey areas),
calculated using the function *loess* in R (R Core Team, 2013). The base and top of the
Nuccaleena Formation show a higher proportion of detrital minerals. Note that this is unlikely
to affect Li isotopic compositions, since they were measured on leaching solutions, which
selectively target the carbonate fraction. All figures were drafted using the *ggplot2* package in
R (Wickham, 2016).
Figure 3. Carbon isotopic compositions in cap dolostones, as a function of the height above the
base of the formation (blue: Nuccaleena, red: Doushantuo). Carbon isotopic compositions were
measured on bulk rock samples. External uncertainty ($2\sigma$) is smaller than the symbol size.
Figure 4. Carbon and oxygen isotopic compositions in the Nuccaleena (blue) and Doushanto
Formation (red) Fms. The Doushanto cap dolostones show a positive correlation between C
and O isotopic compositions ($R^2 = 0.49$; excluding the sample with a $\delta^{13}C > -4$ ‰), suggesting
a possible diagenetic overprint; this is not observed in the Nuccaleena Fm. External uncertainty
($2\sigma$) is smaller than the symbol size.
Figure 5. Lithium isotopic compositions in cap dolostones, as a function of the above the base
of the formation. Lithium isotopic compositions were measured on leaching. The curves are
polynomial fits shown with 95% confidence intervals (grey areas), calculated using the
function *loess* in R and a span value of 0.6. Errors bars show the $2\sigma$ external uncertainty.
Figure 6. Lithium isotopic compositions as a function of Mn/Sr ratios in leaching solutions of
Nuccaleena (blue) and Doushantuo (red) cap dolostones. Errors bars show the $2\sigma$ external
uncertainty.
Figure 7. Lithium isotopic compositions in leaching solutions as a function of oxygen isotopic
compositions of bulk samples in the Nuccaleena (blue) and Doushanto Formation (red) Fms.
The error bar in the top left corner shows the $2\sigma$ external uncertainty on $\delta^7Li$ values.
Figure 8. Calculated seawater Li isotopic compositions for the Nuccaleena Fm, as a function
of the height above the base of the formation. The composition of seawater is calculated using
that of cap dolostone leaching solutions, and the relationship between dolomite and temperature
determined experimentally in Taylor et al. (2018). Temperatures considered for dolostone
formation are 10 (blue symbols), 25 (green) and 40 (red) °C. Curves are polynomial fits shown
with 95% confidence intervals (grey areas), calculated using the function *loess* in R. The grey
areas show the Elatina-Nuccaleena and Nuccaleena-Brachina transition zones.
Figure 9. Modelled seawater Li isotopic compositions (black curves) and calculated seawater
compositions for the Nuccaleena Fm (blue), as a function of the height above the base of each



formation. Seawater compositions were calculated for a temperature of dolostone formation of 25 °C. (a) The hydrological cycle resumes at the onset of cap carbonate deposition (scenario 1; Table 4); (b) the hydrological cycle resumes 1 Myr before the onset of cap carbonate deposition (scenario 2; Table 4).

Figure 10. Modelled seawater Li isotopic compositions (black curves) and calculated seawater compositions for the Nuccaleena Fm (blue), as a function of the height above the base of each formation. Seawater compositions were calculated for a temperature of dolostone formation of 25 °C. For each model curve, during the Marinoan glaciation, the riverine Li supply to the oceans is 80% that of the modern value. (a) The hydrological cycle resumes at the onset of cap carbonate deposition (scenario 4; Table 4); (b) the hydrological cycle resumes 1 Myr before the onset of cap carbonate deposition (scenario 3; Table 4).

Figure 11. Modelled seawater Li isotopic compositions (black curves) and calculated seawater compositions for the Nuccaleena Fm (blue), as a function of the height above the base of each formation. Seawater compositions were calculated for a temperature of dolostone formation of 25 °C. The model curve is calculated assuming a duration of cap dolostone deposition of 0.3 Myr (scenario 5; Table 4).

Figure 12. Modelled seawater Li isotopic compositions (black curves) and calculated seawater compositions for the Nuccaleena Fm (blue), as a function of the height above the base of each formation. Seawater compositions were calculated for a temperature of dolostone formation of 25 °C. The model curve is calculated assuming a stratified ocean during stage 2 (deglaciation) while the ocean is well-mixed during stages 1 (glaciation) and 3 (cap dolostone deposition) (scenario 6; Table 4).





**Tables**

Table 1. Mineral concentrations in bulk rock samples of the Nuccaleena Formation at Elatina Creek

| Sample ID | Height above the base of the formation (m) | Quartz | Aragonite | Albite | Calcite | Dolomite | Ankerite | Siderite | Kaolinite | Chlorite | Illite | Muscovite |
|---|---|---|---|---|---|---|---|---|---|---|---|---|
| EC1 | -0.6 | - | - | - | - | - | - | - | - | - | - | - |
| EC2 | -0.3 | 32 | 1.1 | 11 | 0 | 35 | 0.9 | 0.1 | 2.1 | 2.7 | 0 | 15 |
| EC3 | -0.1 | 20 | 0 | 1.7 | 0.2 | 63 | 3.6 | 1.3 | 1.7 | 1.2 | 0 | 6.6 |
| EC4 | 0.1 | 4.1 | 0 | 0 | 4.6 | 80 | 6.4 | 0.8 | 1.1 | 0 | 0.8 | 2.1 |
| EC5 | 0.2 | 2.6 | 0 | 0 | 3 | 84 | 4.5 | 0.6 | 0.5 | 0 | 0 | 4.8 |
| EC6 | 0.35 | 4.0 | 0.2 | 0 | 7.9 | 74 | 6.2 | 0 | 0.3 | 0 | 0 | 6.6 |
| EC7 | 0.45 | 3.0 | 0.4 | 0 | 2.4 | 83 | 4.8 | 0.4 | 0.5 | 0 | 0 | 4.6 |
| EC8 | 0.45 | 3.6 | 0.1 | 0 | 1.9 | 83 | 6.4 | 0 | 0.4 | 0 | 0.1 | 3.6 |
| EC9 | 0.55 | 4.3 | 0.3 | 0 | 2.6 | 81 | 6.4 | 0.2 | 0.3 | 0 | 0 | 4.6 |
| EC10 | 1.0 | 3.3 | 0.1 | 0 | 0.8 | 79 | 10 | 0.7 | 0.3 | 0 | 1.1 | 4.1 |
| EC11 | 1.2 | 5.9 | 0.4 | 0.1 | 3.7 | 75 | 10.4 | 0.7 | 0.6 | 0 | 1.3 | 1.9 |
| EC12 | 1.35 | 4.3 | 0.3 | 0 | 6.3 | 73 | 5 | 1.4 | 0.9 | 0 | 0 | 8.5 |
| EC13 | 1.4 | 8.7 | 0.5 | 0 | 7.1 | 70 | 6.4 | 0.4 | 0.3 | 0 | 0.3 | 6.1 |
| EC14 | 1.7 | 4.8 | 0.1 | 0 | 5.8 | 74 | 8.2 | 0.4 | 1.4 | 0 | 0 | 5.4 |
| EC15 | 1.8 | 4.2 | 0 | 0 | 11 | 71 | 5.8 | 1.2 | 0.7 | 0 | 0 | 6.3 |
| EC16 | 1.9 | 2.4 | 0.2 | 0 | 0.8 | 79 | 10.8 | 0.6 | 0.3 | 0 | 1.2 | 4.7 |
| EC17 | 2.5 | 3.8 | 0.2 | 0 | 1.7 | 78 | 9.9 | 0.1 | 0.4 | 0 | 0.3 | 5.6 |
| EC18 | 3.0 | 7.2 | 0.3 | 0.3 | 0.8 | 79 | 7.2 | 0.8 | 1.1 | 0 | 0.8 | 2.6 |
| EC19 | 3.4 | 8.1 | 0 | 0.1 | 2.1 | 78 | 7.2 | 0.9 | 0.3 | 0 | 0.9 | 1.8 |
| EC20 | 3.6 | 8.9 | 0 | 0.8 | 1.8 | 78 | 6.3 | 0.4 | 0.4 | 0 | 0.6 | 2.7 |
| EC21 | 3.8 | 9.9 | 0.4 | 1.2 | 0.4 | 77 | 5.6 | 0.9 | 0.6 | 0 | 1.1 | 2.9 |
| EC22 | 3.9 | 6.0 | 0 | 0 | 1.4 | 83 | 5.8 | 0 | 0.7 | 0 | 0.2 | 2.8 |
| EC23 | 4.1 | 8.4 | 0 | 0.9 | 1.2 | 76 | 6.4 | 0.2 | 0.8 | 0 | 0 | 5.5 |
| EC24 | 4.1 | 7.6 | 0 | 1 | 0.3 | 77 | 7.9 | 0 | 0.9 | 0 | 0 | 4.3 |







|  |  |  |  |  |  |  |  |  |  |  |  |  |
| --- | --- | --- | --- | --- | --- | --- | --- | --- | --- | --- | --- | --- |
| EC25 | 4.95 | 8.9 | 0 | 1.5 | 1.2 | 78 | 5.9 | 0 | 0.3 | 0 | 1 | 2.2 |
| EC26 | 5.3 | 14 | 0 | 5.1 | 1.1 | 70 | 2.1 | 0.2 | 1 | 0.2 | 0 | 6.2 |
| EC27 | 5.8 | 16 | 0 | 3.5 | 17 | 44 | 2 | 1.3 | 2.5 | 1.9 | 0 | 12 |
| EC28 | 6.15 | 24 | 1.5 | 7.8 | 0.3 | 33 | 2.6 | 0.9 | 3.1 | 5.7 | 0 | 21 |

Concentrations are given in wt %.




Table 2. Element concentrations, lithium, oxygen and carbon isotopic compositions of the Nuccaleena Formation at Elatina Creek

| Sample ID | Mg (ppm) | Ca (ppm) | Al (ppm) | Ti (ppm) | Mn (ppm) | Rb (ppb) | Sr (ppb) | Li (ppb) | $\delta^7Li$ (‰) | $\delta^{13}C$ | $\delta^{18}O$ | $\delta^7Li_{sw}$ @10°C (‰) | $\delta^7Li_{sw}$ @25°C (‰) | $\delta^7Li_{sw}$ @40°C (‰) |
|---|---|---|---|---|---|---|---|---|---|---|---|---|---|---|
| EC1 | 143 | 125 | 29.4 | 1.45 | 4.23 | 35.8 | 435 | 141 | 2.9 | | | 29.0 | 25.9 | 23.2 |
| EC2 | 288 | 214 | 15.2 | 2.77 | 9.15 | 22.7 | 344 | 44 | 3.1 | -1.06 | -6.87 | 29.2 | 26.1 | 23.4 |
| EC3 | 303 | 220 | 15.6 | 2.89 | 9.92 | 14.6 | 330 | 36 | 4.9 | -1.12 | -7.51 | 31.0 | 27.9 | 25.2 |
| EC4 | 242 | 280 | 30.8 | 3.8 | 2.63 | 8.1 | 572 | 27 | 11.6 | -2 | -8.02 | 37.7 | 34.6 | 31.9 |
| EC5 | 217 | 296 | 17.3 | 3.82 | 1.91 | 8.5 | 405 | 19 | 11.2 | -2.21 | -7.41 | 37.3 | 34.2 | 31.5 |
| EC6 | 191 | 305 | 16 | 3.98 | 0.47 | 12.9 | 524 | 28 | 9.7 | -2.91 | -6.29 | 35.8 | 32.7 | 30.0 |
| EC7 | 233 | 274 | 25.3 | 3.51 | 1.32 | 6.2 | 346 | 22 | 8.1 | -2.13 | -7.68 | 34.2 | 31.1 | 28.4 |
| EC8 | 242 | 268 | 17.1 | 3.37 | 1.82 | 7.7 | 435 | 22 | 10 | -1.73 | -7.89 | 36.1 | 33.0 | 30.3 |
| EC9 | 223 | 277 | 14.7 | 3.56 | 0.98 | 14.2 | 389 | 25 | 11.9 | -1.96 | -7.08 | 38.0 | 34.9 | 32.2 |
| EC10 | 330 | 222 | 16.7 | 2.85 | 0.78 | 11.2 | 285 | 34 | 7.3 | -2.13 | -7.85 | 33.4 | 30.3 | 27.6 |
| EC11 | 210 | 273 | 19.8 | 3.52 | 0.59 | 23.7 | 352 | 39 | 9.4 | -2.49 | -7.34 | 35.5 | 32.4 | 29.7 |
| EC12 | 130 | 351 | 19.9 | 4.57 | 1.77 | 8.1 | 364 | 28 | 11.3 | -2.28 | -7.65 | 37.4 | 34.3 | 31.6 |
| EC13 | 189 | 292 | 22.7 | 3.78 | 3.7 | 14.9 | 294 | 23 | 10.3 | -2.67 | -6.83 | 36.4 | 33.3 | 30.6 |
| EC14 | 138 | 341 | 21.4 | 4.44 | 2.24 | 8.9 | 333 | 32 | 10.8 | -2.89 | -7.73 | 36.9 | 33.8 | 31.1 |
| EC15 | 112 | 355 | 18 | 4.7 | 0.8 | 9.3 | 343 | 25 | 13.7 | -3.19 | -7.28 | 39.8 | 36.7 | 34.0 |
| EC16 | 304 | 239 | 18.7 | 3.14 | 5.41 | 10.2 | 340 | 38 | 9.3 | -2.34 | -8.79 | 35.4 | 32.3 | 29.6 |
| EC17 | 264 | 248 | 17.3 | 3.26 | 5.4 | 8 | 325 | 43 | 10.6 | -2.34 | -8.9 | 36.7 | 33.6 | 30.9 |
| EC18 | 299 | 233 | 26.6 | 2.97 | 5.16 | 7.1 | 273 | 28 | 9.2 | -2.39 | -8.49 | 35.3 | 32.2 | 29.5 |
| EC19 | 229 | 263 | 15.9 | 3.53 | 4.42 | 10.7 | 355 | 27 | 9.7 | -2.57 | -8.28 | 35.8 | 32.7 | 30.0 |
| *replicate* | | | | | | | | | | -2.65 | -8.33 | | | |
| *replicate* | | | | | | | | | | -2.61 | -8.43 | | | |
| EC20 | 293 | 218 | 16.5 | 2.93 | 4.59 | 13 | 262 | 27 | 9.2 | -2.57 | -8.37 | 35.3 | 32.2 | 29.5 |
| EC21 | 279 | 223 | 16.7 | 2.86 | 7.24 | 6 | 287 | 26 | 9.7 | -2.52 | -8.72 | 35.8 | 32.7 | 30.0 |
| EC22 | 264 | 235 | 15.4 | 3.08 | 6.51 | 4.5 | 268 | 21 | 9.4 | -2.49 | -8.05 | 35.5 | 32.4 | 29.7 |
| EC23 | 284 | 223 | 20.1 | 2.86 | 2.88 | 15.2 | 296 | 24 | 10.3 | -2.2 | -8.48 | 36.4 | 33.3 | 30.6 |
| EC24 | 300 | 216 | 16.7 | 2.8 | 9 | 18.3 | 245 | 21 | 8.8 | -2.51 | -8.29 | 34.9 | 31.8 | 29.1 |


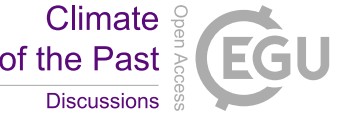



| | | | | | | | | | | | | | |
|---|---|---|---|---|---|---|---|---|---|---|---|---|---|
| EC25 | 269 | 234 | 25.4 | 3.12 | 2.31 | 8.4 | 603 | 34 | 11.7 | -2.92 | -8.06 | 37.8 | 34.7 | 32.0 |
| EC26 | 283 | 218 | 620.4 | 2.89 | 8 | 12.6 | 310 | 20 | 9.6 | -2.94 | -8.23 | 35.7 | 32.6 | 29.9 |
| EC27 | 84 | 343 | 15.6 | 4.56 | 3.13 | 19 | 331 | 28 | 7.2 | | | 33.3 | 30.2 | 27.5 |
| EC28 | 269 | 197 | 15.4 | 2.72 | 11.37 | 22.3 | 340 | 47 | 6.2 | -2.85 | -6.92 | 32.3 | 29.2 | 26.5 |

Element concentrations and Li isotope compositions were measured in leaching solutions while oxygen and carbon isotope compositions were measured in bulk rock. External analytical uncertainty is 1.2 ‰ for $\delta^7Li$, 0.065 ‰ for $\delta^{13}C$ and 0.12 ‰ for $\delta^{18}O$ (2σ). $\delta^7Li_{sw}$ @ 10, 25 and 40°C are the seawater compositions calculated using the relationship between the Li isotope fractionation factor and precipitation temperature from Taylor et al. (2018), and temperatures of cap dolostone formation of 10, 25 and 40 °C.





Table 3. Trace element concentrations, lithium, oxygen and carbon isotopic compositions of the Doushantuo Formation in core 14ZK

| Sample ID | Height above the base of the formation (m) | Mn (ppm) | Rb (ppb) | Sr (ppb) | Li (ppb) | δ7Li (‰) | δ7Li$_{sw}$ @10°C (‰) | δ7Li$_{sw}$ @25°C (‰) | δ7Li$_{sw}$ @40°C (‰) | δ13C | δ18O |
|---|---|---|---|---|---|---|---|---|---|---|---|
| DST-1 | 0.49 | 8.89 | 32.4 | 2328 | 117 | 3.5 | 29.6 | 26.5 | 23.8 | -6.24 | -10.45 |
| DST-2 | 0.64 | 0.09 | <d.l. | 637 | 63 | 4.1 | 30.2 | 27.1 | 24.4 | -4.42 | -9.71 |
| replicate | | | | | | 5.4 | 31.5 | 28.4 | 25.7 | | |
| DST-3 | 0.7 | 0.81 | <d.l. | 1206 | 32 | 7.8 | 33.9 | 30.8 | 28.1 | -5.55 | -10.06 |
| DST-4 | 0.78 | 0.24 | <d.l. | 719 | 37 | 4.2 | 30.3 | 27.2 | 24.5 | -4.24 | -9.23 |
| DST-5 | 0.93 | 2.27 | <d.l. | 790 | 35 | 6.7 | 32.8 | 29.7 | 27.0 | -4.46 | -9.06 |
| DST-6 | 1.08 | 12.29 | 17.9 | 2526 | 64 | 3 | 29.1 | 26.0 | 23.3 | -5.28 | -12.15 |
| DST-7 | 1.23 | 6.65 | <d.l. | 1105 | 38 | 6.7 | 32.8 | 29.7 | 27.0 | -5.35 | -9.25 |
| replicate | | | | | | 6.8 | 32.9 | 29.8 | 27.1 | | |
| DST-8 | 1.38 | 6.45 | <d.l. | 902 | 37 | 8.9 | 35.0 | 31.9 | 29.2 | -3.25 | -6.93 |
| DST-9 | 1.48 | 0.30 | <d.l. | 1262 | 27 | 11.1 | 37.2 | 34.1 | 31.4 | -4.9 | -8.22 |
| DST-10 | 1.63 | 0.42 | <d.l. | 1515 | 19 | 10.9 | 37.0 | 33.9 | 31.2 | -4.81 | -9.03 |
| DST-11 | 1.73 | 0.59 | 1.25 | 1632 | 22 | 9.3 | 35.4 | 32.3 | 29.6 | -4.93 | -9.23 |
| DST-12 | 1.83 | 7.14 | 1.25 | 1649 | 43 | 6.9 | 33.0 | 29.9 | 27.2 | | |
| DST-13 | 1.98 | 0.48 | 23.3 | 1932 | 78 | 5 | 31.1 | 28.0 | 25.3 | | |
| DST-14 | 2.08 | 0.14 | <d.l. | 1679 | 22 | 8 | 34.1 | 31.0 | 28.3 | -5.03 | -10.32 |
| DST-15 | 2.33 | 0.23 | 0.28 | 1972 | 23 | 7 | 33.1 | 30.0 | 27.3 | -4.99 | -10.37 |
| DST-16 | 2.63 | 1.82 | <d.l. | 1025 | 62 | 7.1 | 33.2 | 30.1 | 27.4 | -4.63 | -9.19 |
| DST-17 | 2.83 | 1.01 | <d.l. | 1531 | 26 | 6.5 | 32.6 | 29.5 | 26.8 | -4.89 | -9.82 |
| DST-18 | 2.98 | 0.16 | 3.74 | 2448 | 21 | 13 | 39.1 | 36.0 | 33.3 | -5.08 | -10.32 |
| DST-19 | 3.21 | 2.60 | 1.80 | 1141 | 39 | 9.4 | 35.5 | 32.4 | 29.7 | -5.41 | -9.53 |
| DST-20 | 3.36 | 2.82 | 0.28 | 827 | 38 | 10.5 | 36.6 | 33.5 | 30.8 | -5.35 | -8.72 |
| replicate | | | | | | 8.8 | 34.9 | 31.8 | 29.1 | | |
| DST-21 | 3.51 | 3.06 | <d.l. | 726 | 33 | 6.7 | 32.8 | 29.7 | 27.0 | -4.79 | -8.36 |
| DST-22 | 3.66 | 2.88 | 0.14 | 868 | 29 | 6.6 | 32.7 | 29.6 | 26.9 | -4.83 | -9.4 |
| DST-23 | 3.86 | 0.33 | <d.l. | 1665 | 34 | 7.5 | 33.6 | 30.5 | 27.8 | -5.5 | -10.43 |




| Sample | | | | | | | | | | |
|---|---|---|---|---|---|---|---|---|---|---|
| DST-24 | 4.01 | 0.12 | 1.39 | 2443 | 25 | 7.9 | 34.0 | 30.9 | 28.2 | -5.68 | -11.01 |
| DST-25 | 4.16 | 6.71 | 0.97 | 2211 | 32 | 8.9 | 35.0 | 31.9 | 29.2 | -6.18 | -11.82 |
| DST-26 | 4.34 | 4.67 | 1.25 | 1296 | 57 | 6.4 | 32.5 | 29.4 | 26.7 | -5.91 | -9.72 |
| DST-27 | 4.54 | 0.41 | <d.l. | 1028 | 30 | | | | | -5.63 | -9.63 |
| DST-28 | 4.74 | 3.31 | 0.42 | 772 | 46 | 8 | 34.1 | 31.0 | 28.3 | -5.26 | -8.57 |
| DST-29 | 4.84 | 4.03 | 5.13 | 1549 | 41 | 7.3 | 33.4 | 30.3 | 27.6 | -5.31 | -10.77 |
| DST-30 | 4.99 | 0.24 | <d.l. | 1421 | 35 | 6.1 | 32.2 | 29.1 | 26.4 | -5.15 | -10.59 |
| DST-31 | 5.09 | 0.15 | 1.80 | 841 | 41 | 8 | 34.1 | 31.0 | 28.3 | -5.76 | -9.91 |
| *replicate* | | | | | | | | | | -5.72 | -9.92 |
| *replicate* | | | | | | | | | | -5.7 | -10.02 |

Element concentrations and Li isotope compositions were measured in leaching solutions while oxygen and carbon isotope compositions were measured in bulk rock. External analytical
uncertainty is 1.2 ‰ for $\delta^7$Li, 0.065 ‰ for $\delta^{13}$C and 0.12 ‰ for $\delta^{18}$O (2σ). $\delta^7$Li$_{sw}$ @10, 25 and 40°C are the seawater compositions calculated using the relationship between the Li isotope
fractionation factor and precipitation temperature from Taylor et al. (2018), and temperatures of cap dolostone formation of 10, 25 and 40 °C.





Table 4. Model results

| Scenario # | $T_{lag}$ (Myr) | $T_{carb}$ (Myr) | Stage duration (Myr) | $F_r/ F_{r,mod}$ | $\delta^7 Li_r$ |
|---|---|---|---|---|---|
| 1 | 0 | 3 | 15, 0.1, 2.9 | 0,   50, 5 | 23, 35, 20 |
| 2 | 1 | 3 | 15, 1, 3 | 0,   3, 1 | 23, 35, 23 |
| 3 | 1 | 3 | 15, 1, 3 | 0.8,   3, 1 | 23, 30, 23 |
| 4 | 0 | 3 | 15, 0.1, 2.9 | 0.8, 10, 1 | 23, 35, 23 |
| 5 | 1 | 0.3 | 15, 1, 0.3 | 0,   3, 2 | 23, 35, 35 |
| 6 | 1 | 3 | 15, 1, 3 | 0,   3, 5 | 23, 35, 20 |

$T_{lag}$: Delay between end of glaciation and onset of cap carbonate deposition. $T_{carb}$: Duration of cap carbonation
deposition. $F_r/ F_{r,mod}$: riverine Li flux relative to the modern value (10 Gmol/yr), $\delta^7 Li_r$: riverine Li isotopic
composition. Seawater temperature during cap dolostone deposition: 25 °C. Series of three numbers in the three
rightmost columns represent values for stages 1 to 3. Scenarios 1-5: well-mixed ocean; scenario 6: stratified ocean
during stage 2.



Figure 1

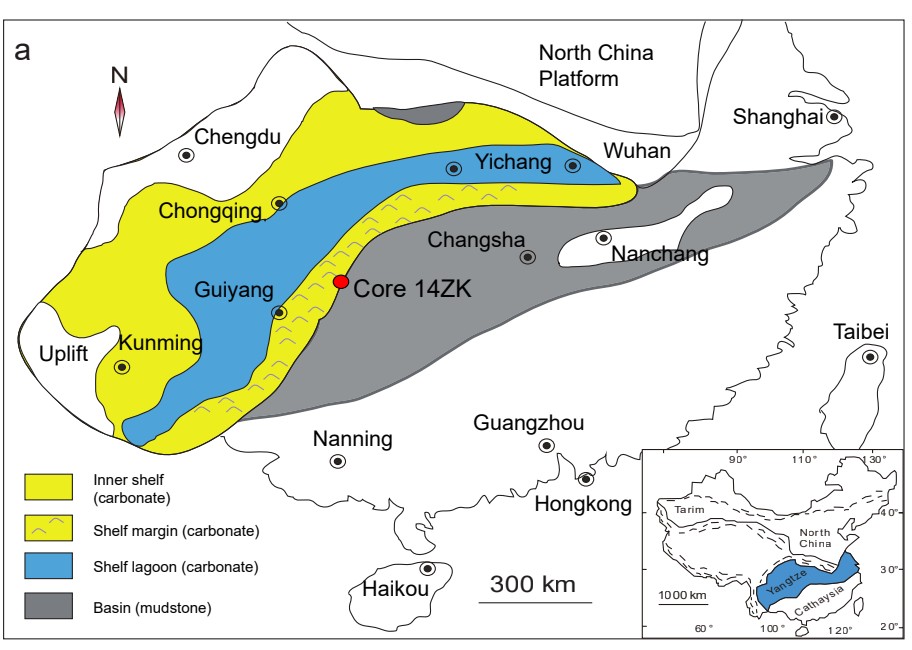

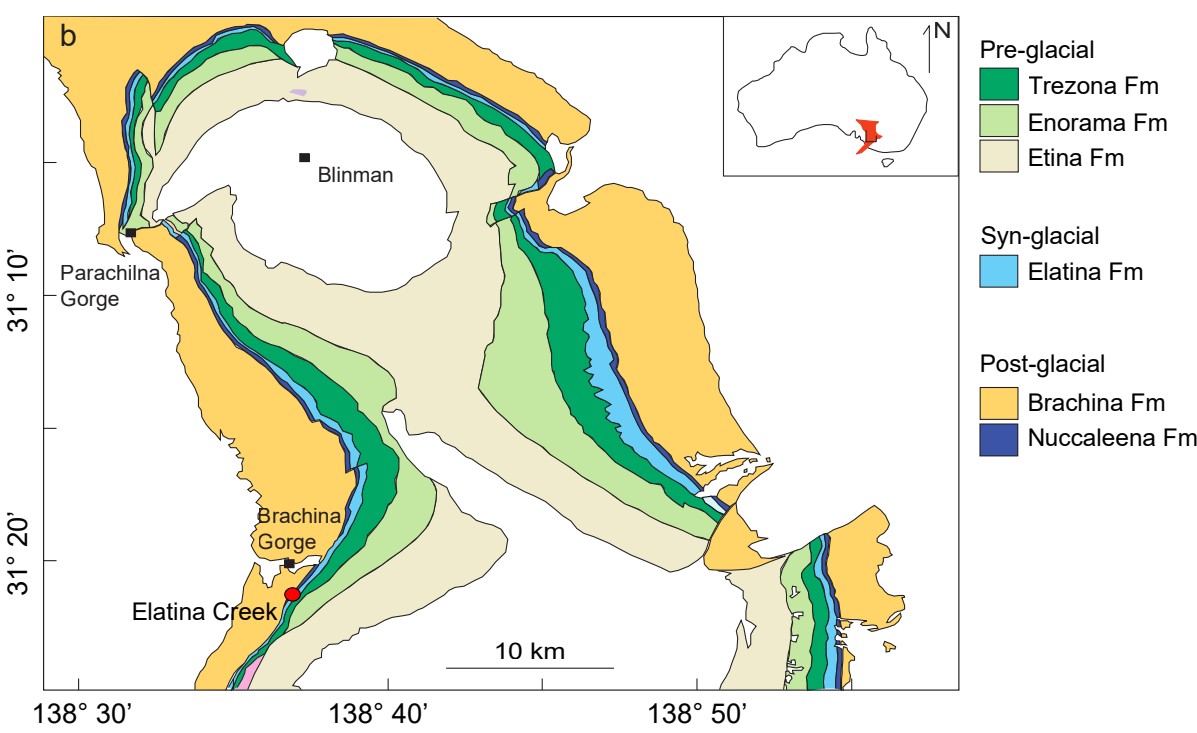



Figure 2

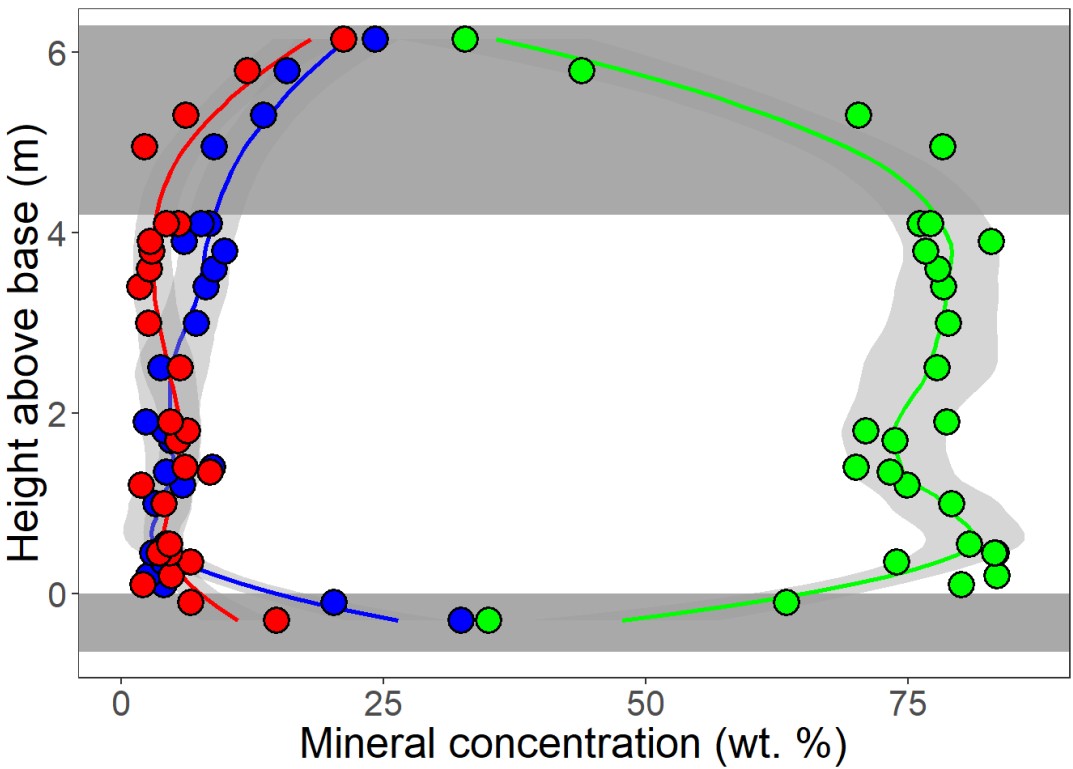





Figure 3

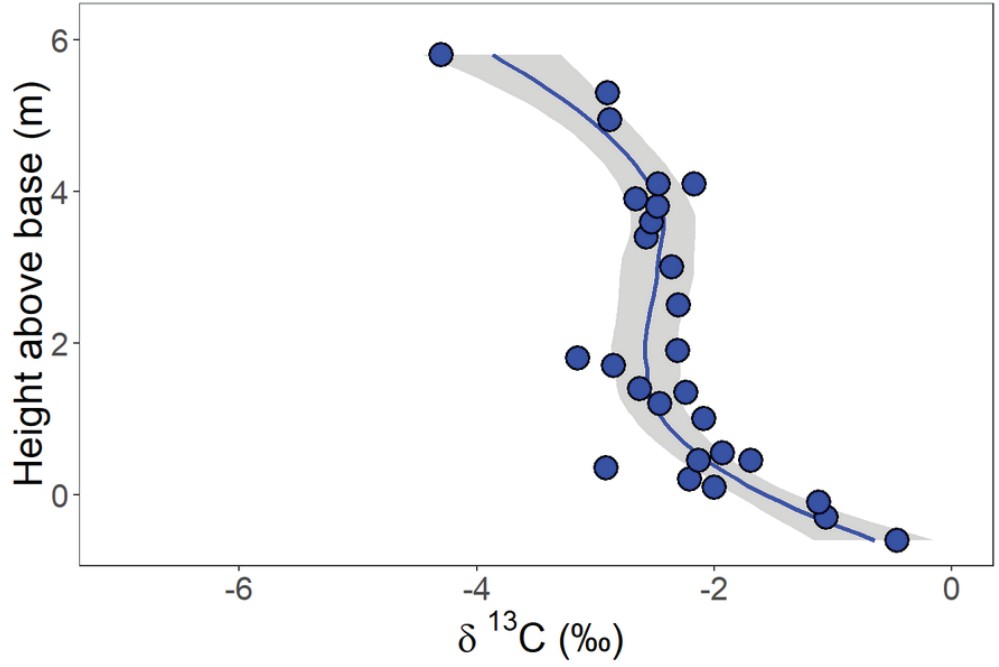

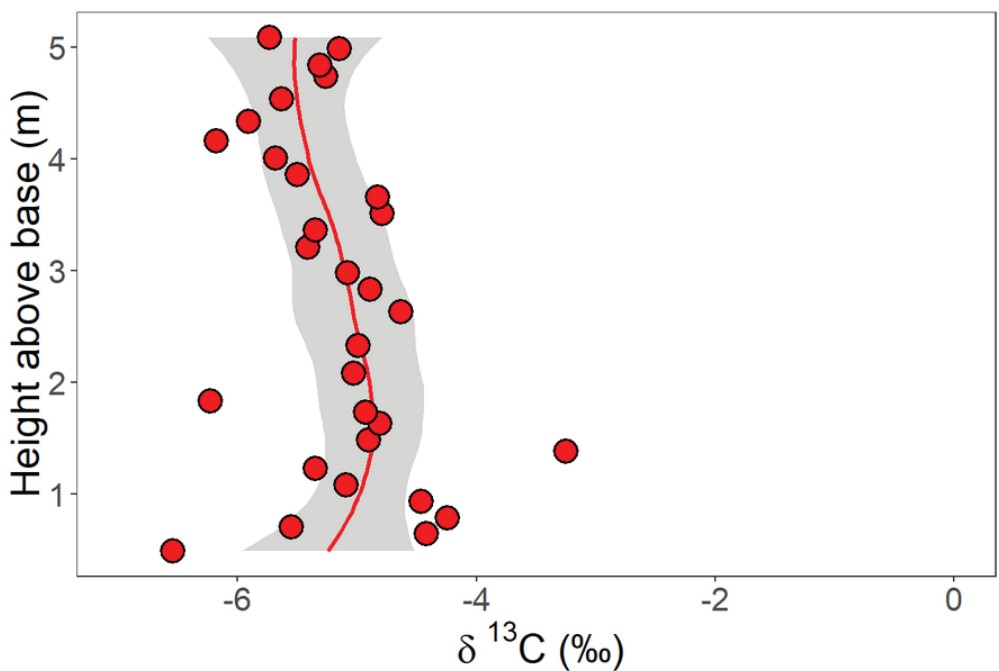





Figure 4

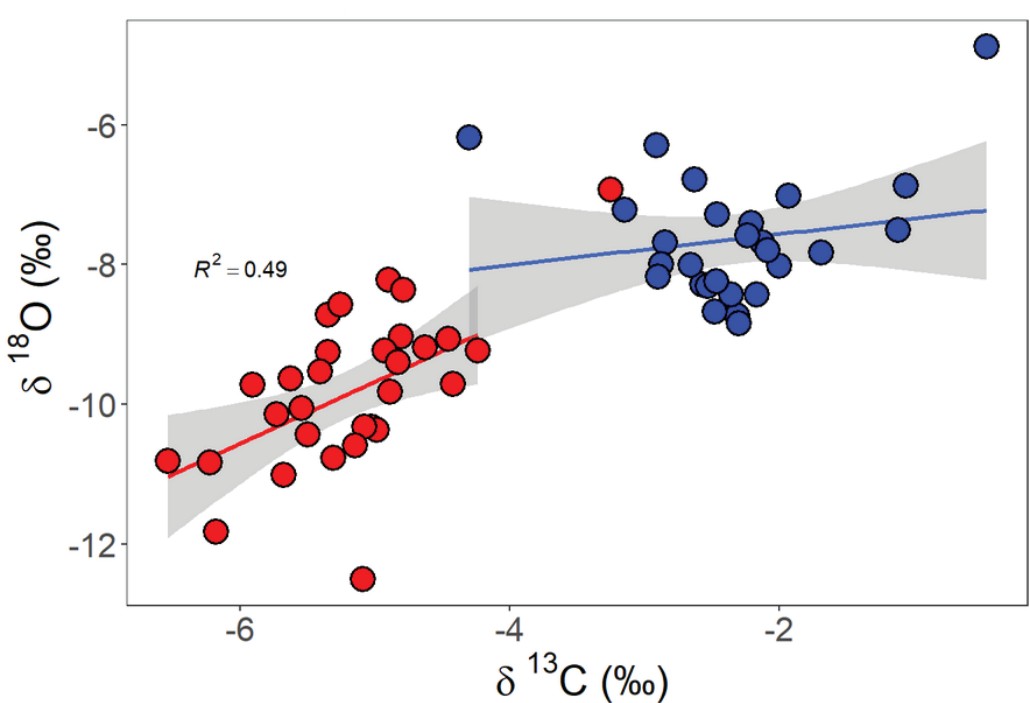



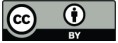

Figure 5

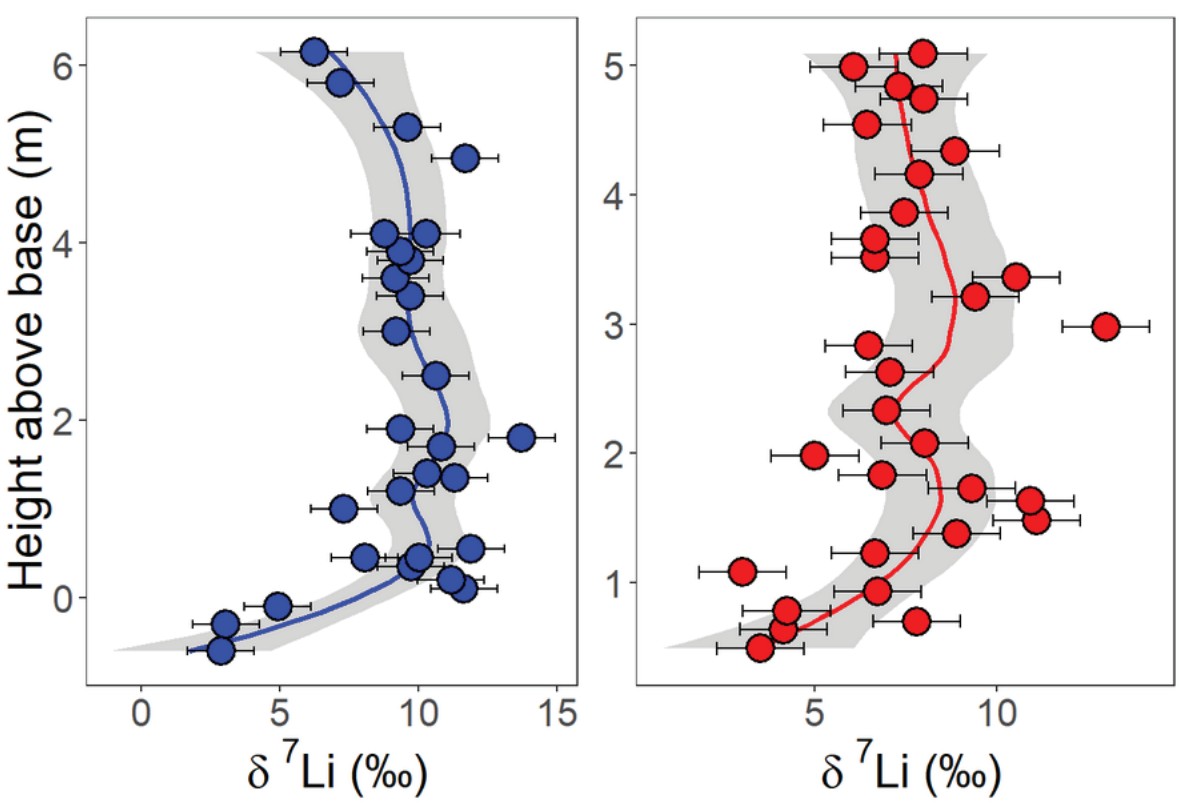





Figure 6

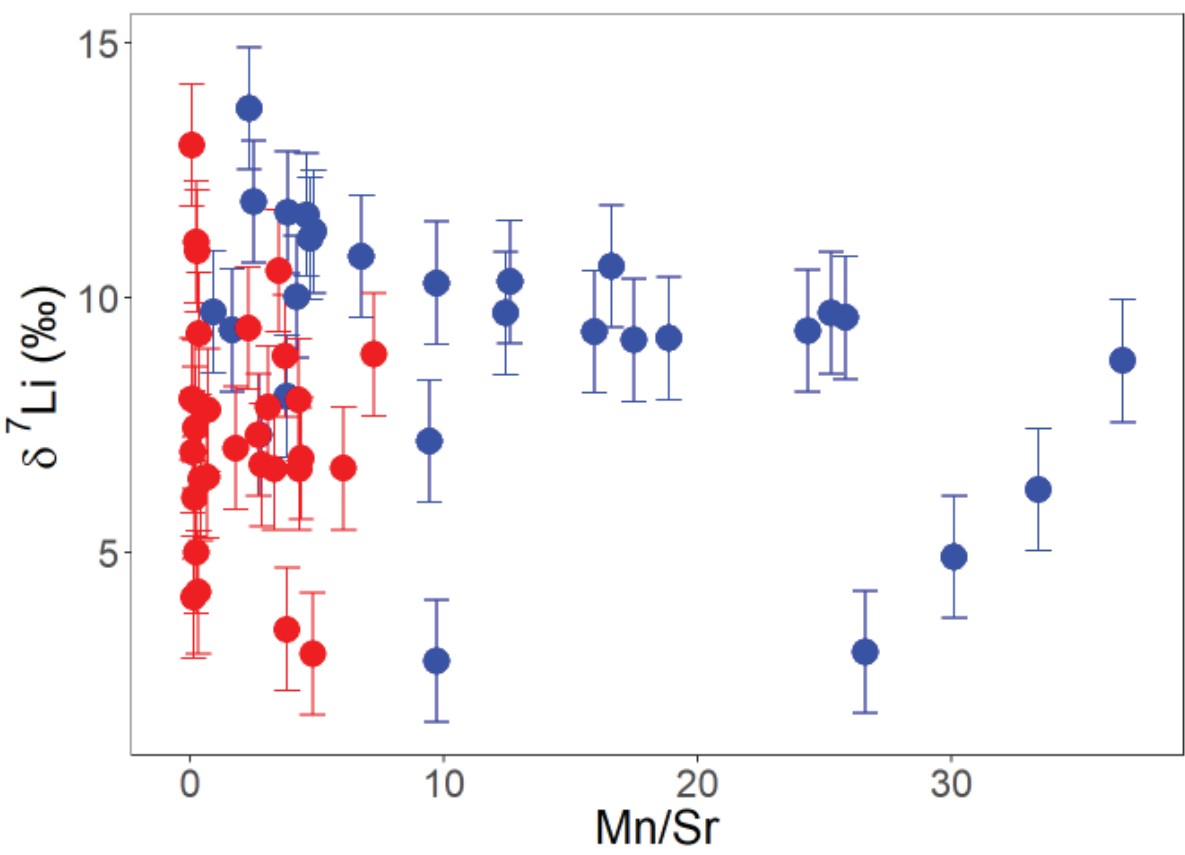





Figure 7

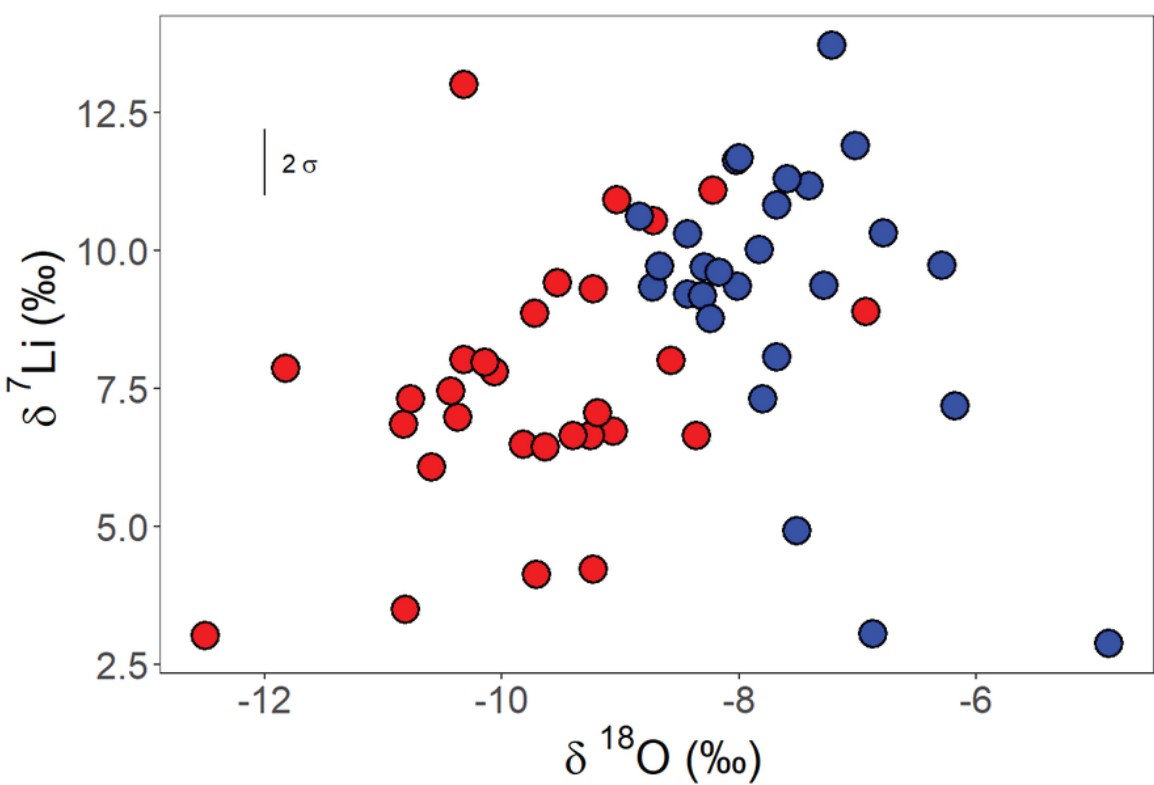




Figure 8

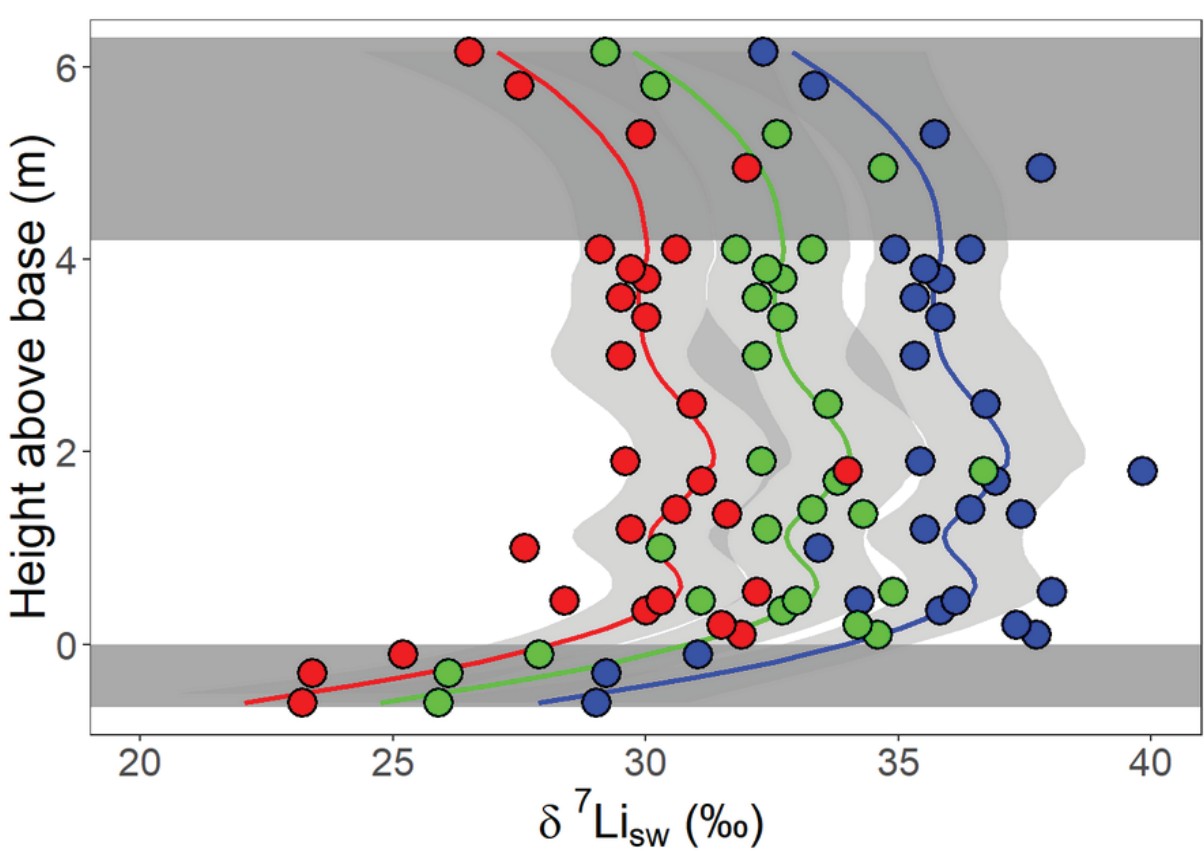




Figure 9







Figure 10

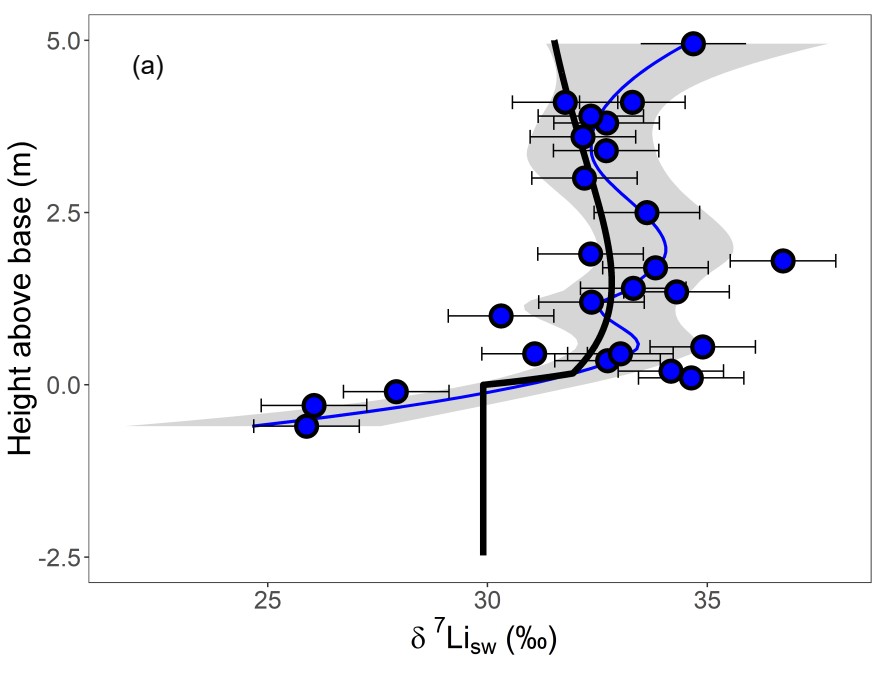

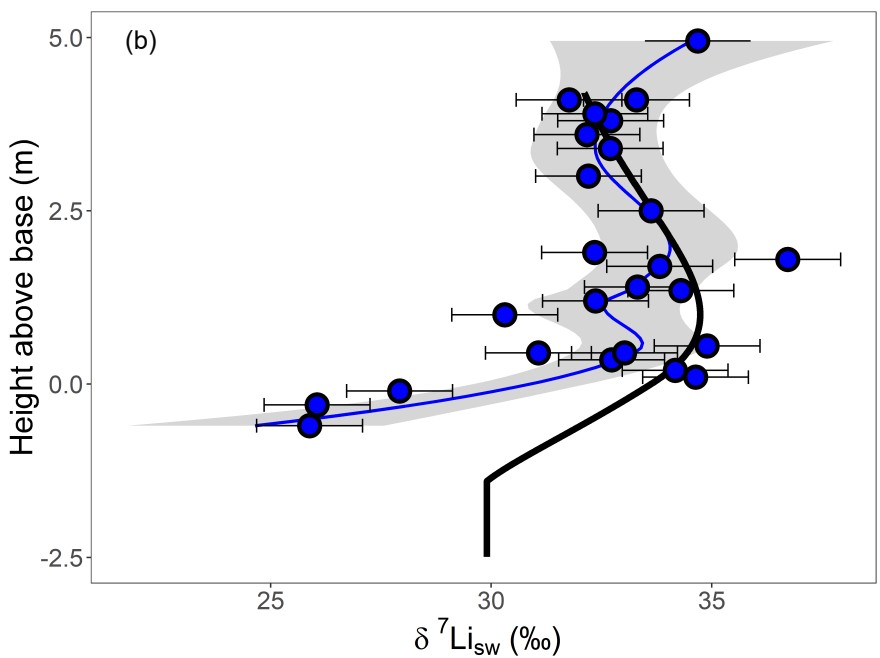





Figure 11

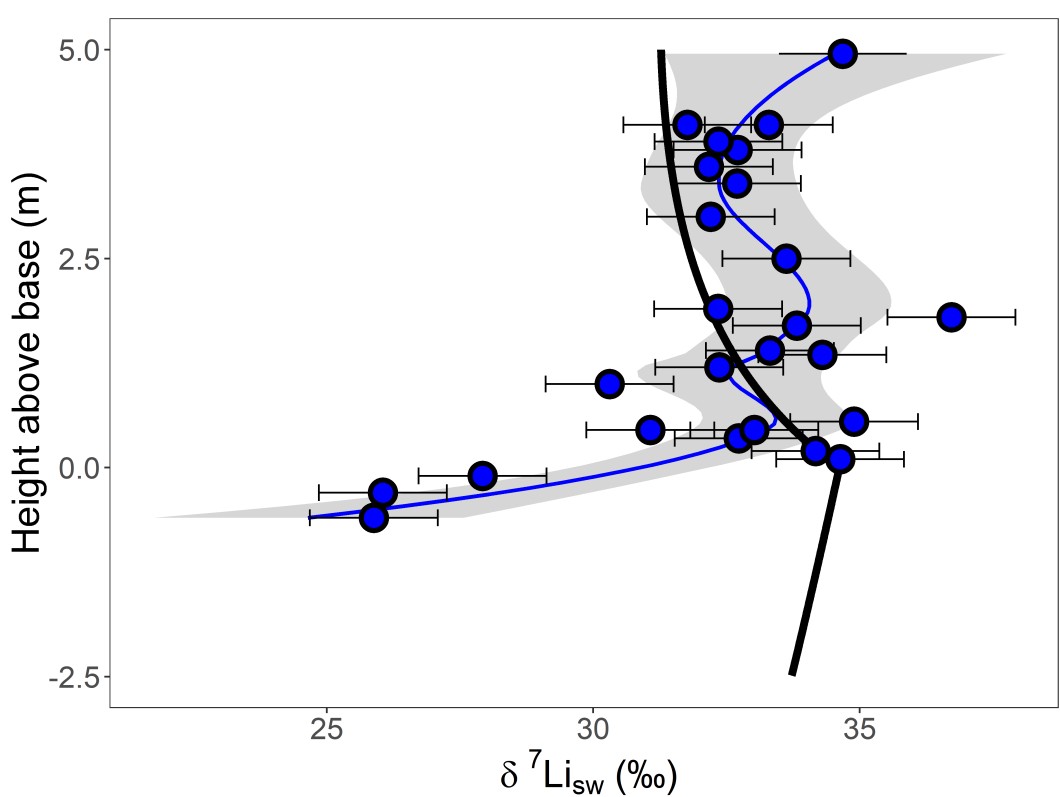




Figure 12

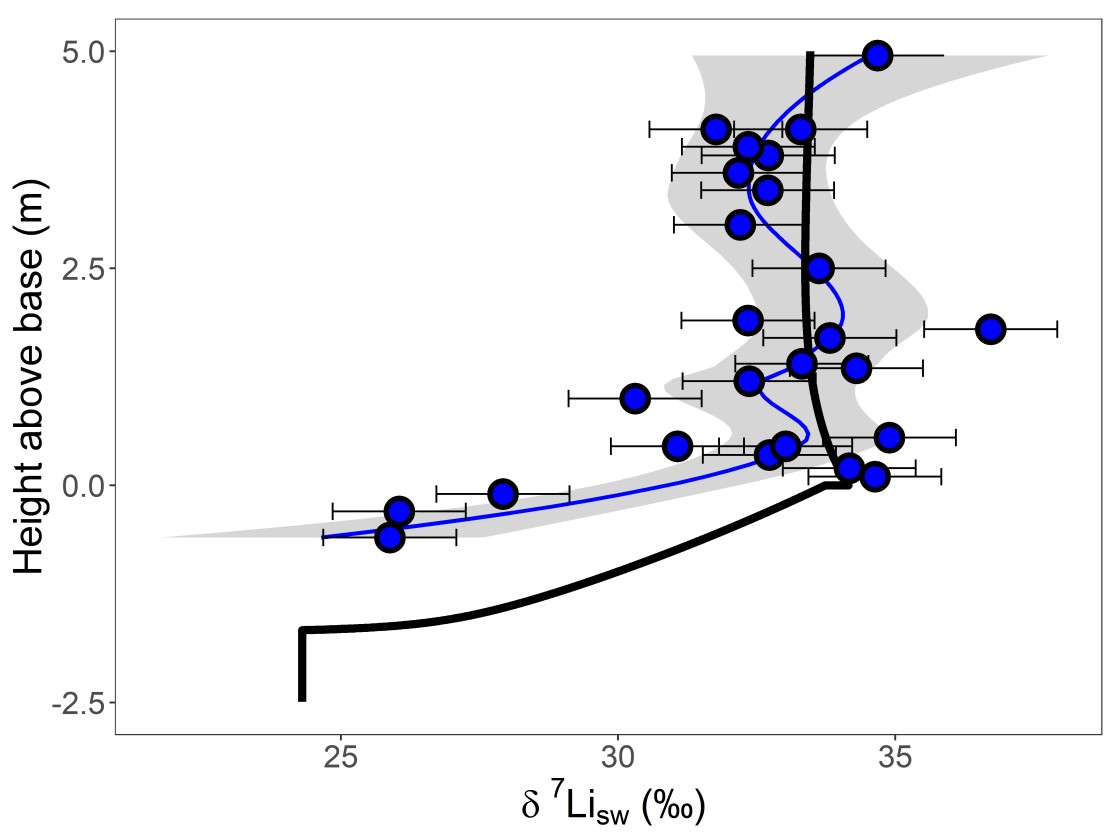