# Peer review of "Rapid recovery of Ediacaran oceans in the aftermath of the"

_Climate of the Past, 2018_

## Referee Comment (RC1) · Anonymous Referee #1 · 8 Jan 2019

The new Li elemental and isotope data is presented for two suites of important past dolostone archives of the Marinoan glaciation/deglaciation (what are the absolute ages of these units anyway?). The manuscript is organized in a logical order although there is a number of figures which could be effectively combined together (Figs 3a,b and Fig 4 could be one two-panel figure, Fig 5 could be reduced to a single panel, and perhaps some of the later figures with models). Then, the difference could be immediately apparent to the reader. I see 12 figures simply as an over-shoot but other than that, the data and interpretation are on a largely sound and reasonable end. What is more serious, and I am sure some of the authors know this fairly well by themselves, the manuscript heavily relies on the results of a study from the same group (Taylor et al. 2018), seemingly submitted to the Earth Surface Dynamics journal in 2018 and stated

as 'pre-print' in the reference list). At least, this is the impression from the manuscript but until today (January 8, 2019), I have not seen this work in that journal nor I obtained any information about the progress of that work. Taking as granted it had happened in 2018, this is a truly sub-optimal behavior, which would normally lead to instantaneous rejection of the manuscript for formal reasons. In this respect, I urge the authors to finish the other experimental manuscript and let it through reviews before this manuscript is accepted. I leave this decision to the Editor-in-Chief. There are some other major issues. For example, the authors discuss diagenetic processes to have affected Doushantuo Fm. samples (L260-263). Diagenesis in carbonates is known to shift d7Li values towards heavier end (see Scholz et al. GCA 2010, You et al. Geophys Res Lett 2003, Ullmann et al. GCA 2013). This could result in modification of authors' observations. Alternatively, they introduce mixing between freshwater and seawater (L263-264). In the latter case, this should be evidenced by some of the key seawater elemental and/or isotope ratios for seawater addition. Considering a key role of these samples suites, I am sure there is a multitude of other elemental data for the samples which could be used. Combined with this, these is some contrast between two places here (L260-263 versus L271). So, either diagenesis had some effect, even if minor, as stated earlier, or it did not have any effect, as suggested later. This should be clarified. To justify dolostones as direct monitors of seawater composition (L289-290) it would require experimental investigation of Li isotope fractionation between a range of natural carbonate chemistry (pure $CaCO_3$, $CaMgCO_3$, $MgCO_3$ etc.) and solutions which is yet missing but would be greatly welcome, considering a possible impact to paleoclimate studies. Therefore, the statement here should be sized down. This is linked to L298 where carbonates are discussed but phase or chemistry of those carbonates is missing. These were likely not dolostones. Please clarify. The serious issue of missing reference and work of Taylor et al. has been discussed above and, therefore, L303 and further, may receive little check in the absence of that paper. Whether the relationship presented in Eq. 1 is true and correct, cannot be evaluated at present. Also, the calculated fractionation factors (L320-321) are burdened with huge uncertainties, so I am not

sure if this has any meaning and if those uncertainties were considered in further calculations. For example, 0.974 +/- 0.241 can easily turn into alpha greater unity and the conclusions would then require a complete re-write. On L325-328, the authors discuss rapid evolution towards modern d7Li values. What are the lines of evidence for this statement? I do not dispute this but some solid piece of evidence should be provided. Residence time of Li is quite long (variably estimated at 1.2-3 Myr), so 'rapid development' may be quite a prolonged period indeed. This is broadly linked with further discussion (L345-346) where constant Li isotope fractionation between sediments and seawater is stated. This is dependent on lithology, in first case. Can we assume that this was similar in Neoproterozoic? Also, was weathering similar before or during onset of Marinoan glaciation similar to modern era? (L348-349) If not, then seawater from that era could be completely different in terms of Li systematics, considering contemporaneous lithological diversity on continents, weathering processes and continental runoff to ocean. Are then the modeled required riverine fluxes real at all? (L371, L374) Is there any other chemical or isotope support, even if from different part of the world? The comparison with modern fluxes (Huh et al. GCA 1998 etc.) may be less straightforward due to landscape and lithology development over time (see also L427-428 where this is discussed; in fact, I am not sure why new rivers would preferentially drain landscapes similar to modern high latitudes. It is the lithology that drives d7Li – Kisakurek et al. Chem Geol 2004, for example). Moreover, majority of modern Icelandic rivers has d7Li at ca. ~20‰ of course with a large range of values. See a summary figure in Tomascak et al. 2016 volume. The oceanic water cycle is assumed to be kept largely similar even during glaciation (L381). I am not sure this is a reasonable expectation. It broadly relates to ocean stratification for one of the models (L406-408) but mixing time of Li is rather fast, on order of ~1 kyr (Misra and Froelich, Science 2012) and such high-resolution data are not available for cap dolostone formation.

Below, some typos and minor issues are listed:

L103: greenhouse stage, not state L134 and 393: several hundred thousand years,

not 100's of kyr L139: . . .this DURATION is significantly shorter than THAT proposed by Condon. . . L163: . . .were used FOR 18O16O. . . L165: is d18O normally expressed relative to NBS19 calcite? Can you rather report it recalculated to V-SMOW? L165: What are the internal errors? 1SD? 2SD? 1SE? 2SE? L166: The same for external REPRODUCIBILITY (not errors) L176: why the bulk and/or silicate fractions were not analyzed? L189: which element(s) was (were) used as internal monitor(s)? L185-196: Flesch et al. have introduced L-SVEC, not IRMM-016. L200: Carignan et al. have not introduced normalization to L-SVEC using MC-ICPMS. The first such measurements were published by Tomascak et al. (Chem Geol 1999). L221-226: this belongs into Discussion, not into Results section. L229: delete (expressed as d7Li). If you want to define this, do it in Section 3.3. L260: avoid using r2 values, anyway this low value of 0.47 is useless and does tell little about any correlation L284-285: is this viable considering a gentle dissolution technique for selective carbonate fraction? It would be hard to believe incomplete isolation of carbonate fraction as suggested on L332-333. Also, Li would unlikely fractionate even if part of carbonate remained intact. L285: more ABUNDANT, not abundance L292: . . .comparing MEAN VALUES for each formation AND excluding. . . L296: . . .probably INDICATING THAT the Li. . . L316: delete OF in the parenthesis L329: replace depending on with DESPITE since temperature is likely not to play any measurable effect on d7Li, as evidenced in several experimental studies L348: put AND between the two references L364: the MODEL results, not model's L365: replace as with BECAUSE L380: SHUTS L398: delete WOULD L424: replace an with Li

---

## Referee Comment (RC2) · Anonymous Referee #2 · 24 Jan 2019

The manuscript presented reports mainly on lithium isotopes in Ediacaran cap dolostones from South Australia (Nuccaleena Formation) and China (Doushantuo Fm) to investigate changes in ocean chemistry that followed the Marinoan deglaciation. In this review, I will focus on the lithium data since they present, from my point of view, the main weakness of this manuscript and do not merit publication in their current form.

The authors are using a new dissolution technique that has been developed and setup by them and is cited as Taylor et al. (2018). This paper is, however, a manuscript or preprint and since September 2018 under review for the journal Climate of the Past and not as yet accepted. In addition, the authors claim that this dissolution or leaching technique is specially set up for dolomites and that only Li from the carbonate/dolomite is leached without any contributions from the siliciclastic detrital component. As such,

the presented data should represent the elemental and lithium isotope composition of the dolomite. While the leaching technique might work for pure carbonates like corals and dolomites, by looking at the results in Table 2, I doubt that this works for impure dolomites. In addition, there seems to be a big discrepancy between the described leaching technique and the results presented.

The authors report that they leached 1 gram of samples with 20 ml (Taylor et al. 2018) of 0.05M HCl for 1 hour. This should dissolve about 5% of the carbonate and should give a rough Ca concentration in the leachate of about 11000 $\mu$g/g. The Ca concentration in the leachates is however on average only 256 ppm. Assuming that ppm here is equivalent to $\mu$g/g, the leachate contained considerably less carbonate, about 0.1% and this doesn't sound right. Admittedly, there are ambiguities associated with the use of 'ppm' but it is not made clear in the manuscript. Alternatively, the authors could have reported the concentration of the 20 ml leachate and this should give 343 mg/L for Ca and 207 mg/L for Mg. In any case, if only material is leached from the dolomite, the Mg/Ca should be around 0.6, but it is instead on average of about 1 and ranges between 0.2 and 1.5. So, something else must have been leached as well. Interestingly, the samples with the lowest Mg/Ca are the ones with the highest amount of calcite (Table 1; note a significant linear correlation between Mg/Ca and the calcite concentration) and the samples with the highest Mg/Ca are the ones with high amounts of the silicate i.e. phyllosilicate minerals (linear correlations between groups of samples and mineral abundances; btw the mineral percentages given in the text line 210 are much higher than what is given in Table 1). That not only dolomite was leached is also obvious from the Al concentration. An average value of 40 ppm looks initially good, but if one assumes that indeed only the dolomite was leached, the Al concentration must have been on average as high as 35000 $\mu$g/g. If we just focus on the ratio, an Al/Ca of 0.16 as given by the data in Table 2 is massive and cannot be leached out of the dolomite alone, most of the Al must have been leached from the silicate component in the sample. Problem is, leaching considerable amounts of aluminium out of silicates results most certainly in leaching lithium as well. Most lithium studies demonstrate

low Al concentrations in the solution that gets analysed for Li, but this is not the case here. Initially, the Li concentration also appears low as expected from pure dolomite, but again, if one compares it with the Ca concentration, the Li concentration would be of 28 $\mu$g/g on average and this is well above everything you can have in a carbonate; the same applies if one is just comparing the Li/Ca ratio. The lithium concentration in carbonates is usually well below 2 $\mu$g/g. If we consider a solution of 5% carbonate in 20 ml, we should find only around 5ppb Li and this is again much lower than the reported average of about 33 ppb in Table 2. If the majority of the Li in the analysed leachate is not from the dolomite, the Li isotope composition cannot represent the composition of the dolomites and can, hence, not be used to investigate changes in ocean chemistry that followed the Marinoan deglaciation.

The manuscript does also not convince that the analytical procedure for lithium is robust. There is vital information missing concerning the e.g. yield and how it was assessed, the procedural blank and most importantly there is no information on the accuracy or on the precision of the Li isotope measurements. How did the authors come up with an "external analytical uncertainty" for the Li isotope values of 1.2 ‰ (especially since it is surprisingly big) and what means $2\sigma$. Regarding the yield, in the referenced Taylor et al. (2018) manuscript it is written that "The columns were calibrated with seawater prior to treating the samples to verify that the procedure yielded 100% of the Li". There are two problems with this. First of all, the seawater data in Table A1 (Column calibration using seawater samples in Taylor et al. (2018)) look not convincing, they scatter between 28.8 and 32.0 ‰.This is a big spread considering that it should be 31.1 ‰ and that seawater has an easy to deal with matrix. Secondly, seawater has a complete different matrix to a carbonate and hence the peak of the Li recovery could be easily shifted. This can only be monitored by taking splits of the solutions collected before and after the column procedure and demonstrating that <0.1% of Li was present in these splits. I would also suggest to not only refer to the paper from Balter and Vigier (2014) but to give a brief description of the method, or rather a detailed description as long as the Taylor et al. (2018) manuscript is not accepted.

The data screen exercise mentioned above can unfortunately only be done for the Australian samples since no mineral concentrations and only limited i.e. insufficient elemental concentrations are given for the Chinese section (Table 3). To assess the purity of the leachate and the validity of the Li data, one needs to have an idea about Ca, Mg and Al concentration. There is, furthermore, no information on the uncertainty of the element concentration data provided. The authors might argue that the mineral and elemental information for the Chinese section is of no interest since the material is anyhow diagentically altered, but I do have doubts about the assessment of the digenetic overprint given in 5.1. Dolomitisation and diagenesis. In general, even if Neoproterozoic carbonate sections are regarded as well-preserved, one still has to visually screen them and this should start already by taking the samples in the field and should be flowed by carefully selecting material that does not contain evidence of secondary alteration or recrystallization usually checked by SEM and CL; has this been done before taking the 1 gr of sample? If yes, that should be documented. To further assess the impact of diagenesis on the sections sampled, the authors (line 257) measured oxygen and carbon isotopic compositions on bulk rock samples of both Nuccaleena and Doushantuo Fms. They report a positive relationship between $\delta18O$ and $\delta13C$ for the Doushantuo Fm of R2= 0.49, but not for the Nuccaleena (Figure 4). If I however, replot the data, I get a much lower R2 and more importantly no statistically significant correlation. Line 265: The effect of diagenesis on Li isotopes was also tested using the Mn/Sr values. I do agree with the authors that an increased Mn/Sr in Neoproterozoic carbonates does not always implies diagenetic alteration, but they forgot to mention in the text that the Mn concentration is unusually high compared to Sr and that the Mn/Sr goes up to 33 (Fig. 5) with only a few samples showing more "normal" ratios. That there is no clear relationship in the leaching solutions between $\delta7Li$ values and Mn/S (Line 268) is not a robust argument that a diagenetic imprint on Mn/Sr has neither a measureable nor a systematic effect on Li isotope compositions. As such I do not see a good argument why the Chinese section is the one that is diagentically altered.

In general, the data are poorly visualized. I am missing a proper stratigraphic section with all isotope and lithological data included. There is also loads of interpretation already in the results section.

---

## Author Comment (AC1) · 18 Mar 2019

We would like to thank Referee 1 for the constructive comments. Each enquiry is addressed below in the order they appeared in the review.

*Referee 1: what are the absolute ages of these units anyway?*

Our response: The age of the cap dolostones is ∼635 Ma (Jiang et a. 2011, Hoffman et al. 2007)

*Referee 1: a number of figures which could be effectively combined together*

[Figure]

Our response: As suggested, figures could be combined together in a revised manuscript.

*Referee 1: the manuscript heavily relies on the results of a study from the same group (Taylor et al. 2018), seemingly submitted to the Earth Surface Dynamics journal in 2018*

Our response: We mistakenly referred to the experimental article on Li isotope fractionation in dolomite as submitted to Earth Surface Dynamics. In fact, it was submitted to Climate of the Past (manuscript cp-2018-113) and since it has been accepted for publication (on 11/03/2019).

*Referee 1: the authors discuss diagenetic processes to have affected Doushantuo Fm. samples (L260-263). Diagenesis in carbonates is known to shift d7Li values towards heavier end (see Scholz et al. GCA 2010, You et al. Geophys Res Lett 2003, Ullmann et al. GCA 2013). This could result in modification of authors' observations.*

Our response: As indicated by Referee 1, diagenesis could shift $\delta^7 Li$ values. In the original manuscript, we indicated that because diagenesis may have affected the Doushantuo Fm (Formation), the Li isotope compositions for this formation are not included in the model discussion. A revised manuscript could emphasize these points and include the references kindly provided by the referee.

*Referee 1: they introduce mixing between freshwater and seawater (L263-264). In the latter case, this should be evidenced by some of the key seawater elemental and/or isotope ratios for seawater addition. Considering a key role of these samples suites, I am sure there is a multitude of other elemental data for the samples which could be used.*

Our response: The discussion of freshwater and seawater mixing (the *plumewater* model of Shields, 2005) is not intended to be a test of the validity of this model (this was done in other publications); but only to test its incidence on our Li data since it has been proposed as a possible model for cap carbonate formation.

*Referee 1: either diagenesis had some effect, even if minor, as stated earlier, or it did not have any effect, as suggested later. This should be clarified.*

Our response: There seems to be some confusion about whether diagenesis had some effect or not. As indicated in the original manuscript, we propose it may have had an effect on the Doushantuo Fm but there is no evidence that it is the case for the Nuccaleena Fm.

*Referee 1: experimental investigation of Li isotope fractionation between a range of natural carbonate chemistry (pure CaCO3, CaMgCO3, MgCO3 etc.) and solutions which is yet missing but would be greatly welcome, considering a possible impact to paleoclimate studies.*

Our response: Experimental investigation of Li isotope fractionation between a range of natural carbonate chemistry (pure $CaCO_3$, $CaMgCO_3$, $MgCO_3$ etc.) has been done for calcium carbonate (Marriott et al. 2004a, b), and results for Ca-Mg and Mg carbonates are presented in Taylor et al. (2019) (cp-2018-113).

*Referee 1: phase or chemistry of those carbonates is missing. These were likely not dolostones. Please clarify.*

Our response: The rocks studies are indeed dolostones, please refer to Section 2 and references therein in the original manuscript.

*Referee 1: calculated fractionation factors (L320-321) are burdened with huge uncertainties*

Our response: The concern about the uncertainty on the isotopic fractionation factor is a fair point. It would be more appropriate to report the difference of isotope ratios between mineral and solution ($\Delta^7 Li_{min-sol}$), rather that the isotopic fractionation factor ($\alpha_{min-sol}$). In this case, we can see that seawater temperatures of 10 and 40 °C give ($\Delta^7 Li_{min-sol}$) values of -26 $\pm$ 9 ‰and -20 $\pm$ 7 ‰, respectively (thus seawater being between 26 $\pm$ 9 ‰and 20 $\pm$ 7 ‰higher than dolomite). These values are also within error of each other, and prevents the issue of an isotopic fractionation factor becoming greater than 1 (and thus an opposite isotopic fractionation).

*Referee 1: On L325-328, the authors discuss rapid evolution towards modern d7Li values. What are the lines of evidence for this statement? I do not dispute this but some solid piece of evidence should be provided. Residence time of Li is quite long (variably estimated at 1.2-3 Myr), so 'rapid development' may be quite a prolonged period indeed.*

Our response: The lines of evidence for a rapid evolution towards modern $\delta^7 Li$ values are the data themselves and the previously proposed duration for cap carbonates deposition (Condon et al., 2005). Considering the latter, the increase in $\delta^7 Li$ values at the base of the formation towards modern values would suggest this change occurs in as little as 0.1 Myr.

*Referee 1: constant Li isotope fractionation between sediments and seawater is stated. This is dependent on lithology, in first case. Can we assume that this was similar in Neoproterozoic?*

Our response: It is of course difficult to verify whether the Li isotope fractionation between sediments and seawater was constant during the Neoproterozoic. The

same applies to studies of Phanerozoic carbonates, where such constant isotopic fractionation could not be verified but yet was assumed (Lechler et al. 2015, Pogge von Strandmann et al. 2013, 2017).

*Referee 1: was weathering similar before or during onset of Marinoan glaciation similar to modern era?*

Our response: we are indeed unable to test whether weathering was similar before or during onset of Marinoan glaciation similar to modern era. What the data and model show is that shortly after the onset of cap carbonate deposition, the seawater $\delta^7 Li$ returns to values similar to that pre-Marinoan glaciation. The narrative of the manuscript could be changed to reflect this cautious analysis, without impacting the importance of the findings.

*Referee 1: Are then the modeled required riverine fluxes real at all? (L371, L374) Is there any other chemical or isotope support, even if from different part of the world?*

Our response: As indicated in the original manuscript, Kasemann et al. (2005) have also studied the Nuccaleena Fm and showed that Ca supply to the oceans could have been 14 to 140 times greater than the modern flux. Our estimate falls within this range. Note that, as pointed out in the original manuscript, Li isotopes are a more unequivocal proxy for chemical weathering than Ca isotopes (which are also be sensitive to the environment of carbonate formation; e.g. Fantle and Higgins, 2014).

*Referee 1: I am not sure why new rivers would preferentially drain landscapes similar to modern high latitudes*

Our response: Data show that to reproduce the Li isotope composition of oceans at the start of deglaciation, rivers would need to have the same Li isotope composition as

modern high latitude rivers. This does not imply that only high latitudes were drained, but that the average world river would have had a Li isotope composition similar to some of the rivers draining modern high latitude landscapes.

*Referee 1: It is the lithology that drives d7Li – Kisakurek et al. Chem Geol 2004, for example.*

Our response: We respectfully disagree. Lithology imparts a small range of $\delta^7 Li$ values compared to the effect of clay formation (Burton and Vigier, 2011). This is why Li isotopes are such a useful tool to study chemical weathering (the nature of the parent material doesn't matter so much).

*Referee 1: The oceanic water cycle is assumed to be kept largely similar even during glaciation (L381). I am not sure this is a reasonable expectation.*

Our response: Our discussion does not intend to evaluate whether this is a reasonable hypothesis. It is just an end-member model considered to account for a *Slushball* model (i.e. partial glaciation) that others have proposed as an alternative to Snowball Earth (Fairchild and Kennedy, 2007). We just show that even in the extreme scenario where the Slushball was very slushy (i.e. hydrological cycle similar to present), we obtain conclusions similar to when considering a completely glaciated Earth.

*Referee 1: mixing time of Li is rather fast, on order of 1 kyr (Misra and Froelich, Science 2012) and such high-resolution data are not available for cap dolostone formation*

Our response: We agree such high-resolution data are not available. However, considering that the *plumewater* model is one of the plausible models for cap carbonate formation, we thought it was worth testing.

Typos and minor issues kindly identified by Referee 1 could be easily dealt with in a revised manuscript.

**References**

Burton, K.W. and Vigier, N.: Lithium Isotopes as Tracers in Marine and Terrestrial Environments, pp. 41–59, Advances in Isotope Geochemistry, Springer Berlin Heidelberg, 2011.

Condon, D., Zhu, M., Bowring, S., Wang, W., Yang, A., and Jin, Y.: U-Pb ages from the neoproterozoic Doushantuo Formation, China, 5 Science, 308, 95–98, 2005.

Fairchild, I. J. and Kennedy, M. J.: Neoproterozoic glaciation in the Earth System, Journal of the Geological Society, 164, 895–921, 2007.

Hoffman, P. F., Halverson, G. P., Domack, E. W., Husson, J. M., Higgins, J. A., and Schrag, D. P.: Are basal Ediacaran (635 Ma) postglacial "cap dolostones" diachronous?, Earth and Planetary Science Letters, 258, 114–131, 2007.

Jiang, G., Shi, X., Zhang, S.,Wang, Y., and Xiao, S.: Stratigraphy and paleogeography of the Ediacaran Doushantuo Formation (ca. 635–551 Ma) in south China, Gondwana Research, 19, 831–849, 2011.

Kasemann, S. A., Hawkesworth, C. J., Prave, A. R., Fallick, A. E., and Pearson, P. N.: Boron and calcium isotope composition in Neoproterozoic carbonate rocks from Namibia: Evidence for extreme environmental change, Earth and Planetary Science Letters, 231, 73–86, 2005.

Kisakürek, B., Widdowson, M., and James, R. H.: Behaviour of Li isotopes during continental weathering: the Bidar laterite profile, India, Chemical Geology, 212, 27–44, 2004.

Lechler, M., Pogge von Strandmann, P. A. E., Jenkyns, H. C., Prosser, G., and Parente, M.: Lithium-isotope evidence for enhanced silicate weathering during OAE 1a (Early Aptian Selli event), Earth and Planetary Science Letters, 432, 210– 222, 2015.

Marriott, C. S., Henderson, G. M., Belshaw, N. S., and Tudhope, A. W.: Temperature dependence of $\delta^7 Li$, $\delta^{44} Ca$ and Li/Ca during growth of calcium carbonate, Earth and Planetary Science Letters, 222, 615–624, 2004a.

Marriott, C. S., Henderson, G. M., Crompton, R., Staubwasser, M., and Shaw, S.: Effect of mineralogy, salinity, and temperature on Li/Ca and Li isotope composition of calcium carbonate, Chemical Geology, 212, 5–15, 2004b.

Misra, S. and Froelich, P. N.: Lithium isotope history of Cenozoic seawater: Changes in silicate weathering and reverse weathering, Science, 335, 818–823, 2012.

Pogge von Strandmann, P. A. E., Jenkyns, H. C., and Woodfine, R. G.: Lithium isotope evidence for enhanced weathering during Oceanic Anoxic Event 2, Nature Geosci, 6, 668–672, 2013.

Pogge von Strandmann, P. A. E., Desrochers, A., Murphy, M. J., Finlay, A. J., Selby, D., and Lenton, T. M.: Global climate stabilisation by chemical weathering during the Hirnantian glaciation, Geochemical Perspectives Letters, pp. 230–237, 2017.

Shields, G. A.: Neoproterozoic cap carbonates: a critical appraisal of existing models and the plumeworld hypothesis, Terra Nova, 17, 299–310, 2005.

Taylor, H., Kell Duivestein, I., Farkas, J., Dietzel, M., and Dosseto, A.: Lithium isotopes in dolostone as a palaeo-environmental proxy – An experimental approach, Climate of the Past, 2019.

---

## Author Comment (AC2) · 28 Mar 2019

We would like to thank Referee 2 for the constructive comments. Each enquiry is addressed below in the order they appeared in the review.

From Referee's comment C1 to the penultimate paragraph of comment C3, these comments in essence refer to Taylor et al. (2019). Although we acknowledge the difficult job presented to the referee since at the time Taylor et al. (2019) was not accepted, the comments do not directly refer to the present manuscript. Since, Taylor et al. (2019) has been accepted for publication in Climate of the Past (as of 11 March 2019), and trusting Copernicus' peer-review system we have to consider the results in Taylor et al.

[Figure]

(2019) as scientifically sound.

Nevertheless, it is possible to provide evidences that support the applicability of the protocol in Taylor et al. (2019) to this study: in our dataset, there is no relationship between $\delta^7 Li$ and several proxies for silicate contamination in the leaching solution: Li/Ca, Al/Ca or Al/Mg (see Fig .1 for the Nuccaleena Formation). While the absence of evidence is not evidence of absence, and while silicates may be leached with carbonates during sample preparation, if silicates dominated the $\delta^7 Li$ of the leaching solution (as they would if present in significant amount, since Li is much more concentrated in silicates compared to carbonates), we should observe a relationship between $\delta^7 Li$ and Li/Ca, Al/Ca or Al/Mg. This is not observed.

Although samples with Li/Ca $> 20\text{x}10^{-5}$ also show low $\delta^7 Li$ values, there is no systematic variation of the Li/Ca ratio with height above the base of the formation (Fig. 1). Thus, the increase in $\delta^7 Li$ values at the base of the formation is not explained by high Li/Ca (and thus silicate contribution). Even if we rule out samples with Li/Ca $> 20\text{x}10^{-5}$, the conclusions of the study remain: there is a rapid increase in $\delta^7 Li$ values at the base of the formation to go from low modelled $\delta^7 Li$ values at the end of the Marinoan glaciation to the values observed in samples with Li/Ca $< 20\text{x}10^{-5}$.

In regard to information about quality control of Li isotope data (yield and how it was assessed, procedural blank and accuracy or precision), the yield was assessed during column calibration and it was verified to be 100% (which is critical for accurate Li isotope measurements). Total procedure blanks were routinely measured and yielded 2 ng of Li. This represents a contribution to the Li isotope ratio of samples of 0.03 ‰. Accuracy could only be assessed using seawater (see below), since we do not have access to a carbonate reference material (which would have been ideal). While Bastian et al. (2018) recently provided recommended values for calcium carbonate standards Jcp-1 and Jct-1, these materials are not commercially available anymore (and most labs are unwilling to share their stock, which is limited). However,

measurement of a Holocene *Porites* coral yielded a $\delta^7 Li$ of 20.6 ‰, which is similar to values reported for modern *Porites* in Marriott et al. (2004) (between $18.4 \pm 0.4$ and $19.1 \pm 0.4$ ‰). Measurements on a 204,000yr-old *Favia* coral from Cook Islands also yielded a $\delta^7 Li$ of $20.5 \pm 0.5$ ‰(2 SE; n=4). These details can be included to a revised manuscript.

*Referee 2: How did the authors come up with an "external analytical uncertainty" for the Li isotope values of 1.2 ‰ (especially since it is surprisingly big) and what means $2\sigma$.*

Our response: The external analytical uncertainty was calculated by processing separate aliquots of the same sample through the entire process (leaching, chromatography, analysis). This was done for several samples. This error is indeed greater than errors reported when replicating samples that undergo total dissolution (for instance, in our lab we obtain a $2\sigma$ error of 0.2 ‰for silicate reference material JG-2 which undergoes total dissolution), and may be attributed to the leaching step. Note that 1.2 ‰is small compared the range of values observed in the dataset (about 10 ‰). $2\sigma$ refers to 2 standard deviations.

*Referee 2: the seawater data in Table A1 (Column calibration using seawater samples in Taylor et al. (2018)) look not convincing, they scatter between 28.8 and 32.0 ‰This is a big spread considering that it should be 31.1 ‰ and that seawater has an easy to deal with matrix.*

Our response: Note that:

1. The range of values we report in Taylor et al. (2019) (28.8-32.0 ‰) is similar to that in the literature (28.0-32.5 ‰; Pistiner and Henderson, 2003; Carignan et al., 2007). Furthermore, this is only a 1.3 ‰deviation from the accepted value, which is similar to our external uncertainty (see above) and small compared to

the range of variation observed in the samples (about 10 ‰).

2. Seawater is actually a more difficult matrix because of the abundance of Na, which is critical to remove for Li isotope measurement: Na/Li is about 100 times greater in seawater than in carbonates (this is also the case for dolostones, e.g. Fritz and Katz, 1972; Land and Hoops, 1973).

*Referee 2: seawater has a complete different matrix to a carbonate and hence the peak of the Li recovery could be easily shifted.*

Our response: Since Referee 2's comments were received, we have produced an elution curve using a dissolved coral (i.e. calcium carbonate). The elution curve is similar to that produced using seawater (Fig. 2). There is a small shift in Li elution, however this is accounted for by the chosen Li elution volumes. While it seems Li and Na peaks are closer with a carbonate matrix, as pointed out above, the Na/Li ratio of carbonates is about 100 times lower than that of seawater. Thus, even if a greater proportion of the Na peak was collected with the Li elution for carbonates, this still results in a negligible fraction of Na in the final Li elution (i.e. after two passes on the column). For seawater, after two passes on the column, 0.4-0.6 % of Na remain in the Li elution. The elution curve for carbonates show there's 1.6 % of Na left in the Li elution after two passes. Considering that (i) the Na/Li of carbonates is about 100 times lower than in seawater and (ii) we always load the same amount of Li on the column (about 60 ng), after two column passes the amount of Na left in the Li elution for carbonates would be about 30 times lower than that when processing seawater.

*Referee 2: give a brief description of the method*

Our response: The methods given in the original manuscript are indeed too succint, and more details would be given in a revised manuscript, as those presented above.

[Figure]

*Referee 2: The data screen exercise mentioned above can unfortunately only be done for the Australian samples since no mineral concentrations and only limited i.e. insufficient elemental concentrations are given for the Chinese section*

Our response: This is indeed correct. Nevertheless, in the original manuscript we discuss that the Chinese section may have been affected by diagenesis (using C and O isotopes) and thus these date are not used in the model presented in the discussion.

*Referee 2: no information on the uncertainty of the element concentration data provided.*

Our response: precision on element concentration has been assessed by replicating a sample through the whole process (leaching, chromatography and analysis). It is better than 1% for Li, Mg and Ca, 1.5% for Mn and Rb, 6% for Ti and 9% for Al. Error bars are shown accordingly in Fig. 1 (within the symbol size if not shown).

*Referee 2: In general, even if Neoproterozoic carbonate sections are regarded as well-preserved, one still has to visually screen them and this should start already by taking the samples in the field and should be flowed by carefully selecting material that does not contain evidence of secondary alteration or recrystallization usually checked by SEM and CL; has this been done before taking the 1 gr of sample? If yes, that should be documented*

Our response: Samples of the Nuccaleena Fm were taken from the same section in the field as the one sampled by Rose and Maloof (2010), Kunzmann et al. (2013) and Liu et al. (2014). In these studies, they all propose that dolomite is primary and argue against a diagenetic overprint. As indicated in the original manuscript, Hoffmann et al. (2007) indicated that dolomitized carbonates are commonly coarsely recrystallised and occur at the top of shoaling sequences, while the Nuccaleena Fm is fine-grained and was deposited at the base of a transgressive-regressive

sequence. Since we received the Referees' comments, we have undertaken SEM-EDS analysis on selected samples. They typically show fine-grained dolomite, as observed by authors cited above at the section sampled, and where calcite is detected it is in veins cutting and thus post-dating the dolomite, not the other way around (Fig. 3).

*Referee 2: To further assess the impact of diagenesis on the sections sampled, the authors (line 257) measured oxygen and carbon isotopic compositions on bulk rock samples of both Nuccaleena and Doushantuo Fms. They report a positive relationship between $\delta^{18}O$ and $\delta^{13}C$ for the Doushantuo Fm of $R^2$ = 0.49, but not for the Nuccaleena (Figure 4). If I however, replot the data, I get a much lower $R^2$ and more importantly no statistically significant correlation.*

Our response: As Referee 2 points out, this can't be thoroughly tested because we don't have mineralogical data for these samples. Note that if one suggested that the Doushantuo Fm data are also a useful proxy for Neoproterozoic ocean composition (like the Nuccaleena Fm), Li isotope variations in the Doushantuo Fm would show agreement with those in the Nuccaleena Fm, suggesting a global signal. In the original manuscript, we are using caution as not to draw this inference unless entirely confident.

*Referee 2: I do agree with the authors that an increased Mn/Sr in Neoproterozoic carbonates does not always implies diagenetic alteration, but they forgot to mention in the text that the Mn concentration is unusually high compared to Sr and that the Mn/Sr goes up to 33 (Fig. 5) with only a few samples showing more "normal" ratios. That there is no clear relationship in the leaching solutions between $\delta^7 Li$ values and Mn/Sr (Line 268) is not a robust argument that a diagenetic imprint on Mn/Sr has neither a measureable nor a systematic effect on Li isotope compositions. As such I do not see a good argument why the Chinese section is the one that is diagentically altered.*

Our response: There is no clear relationship in the leaching solutions between $\delta^7 Li$

values and Mn/Sr, which is a good proxy for diagenesis as Referee 2 acknowledges. Furthermore, there is also no systematic change in $\delta^7 Li$ for samples that have high concentrations of silicate minerals and high Mn/Sr ratios: some have a low $\delta^7 Li$ , others a high $\delta^7 Li$ . Thus, there is no evidence for a systematic effect of (i) diagenesis or (ii) silicate mineral contribution on $\delta^7 Li$ . It does not imply it is not there, but it is just not significant enough for the range of $\delta^7 Li$ values we measure (about 10 ‰).

*Referee 2: the data are poorly visualized.*

Our response: This can be fixed as suggested by Referee 1 (merging figures into panels).

*Referee 2: I am missing a proper stratigraphic section with all isotope and lithological data included.*

Our response: This can be added, but it is of little value since there is only one lithological unit (stratigraphic sections are more useful when there are several units). If the editor feels this is important, such stratigraphic section can be included to a revised manuscript.

*Referee 2: There is also loads of interpretation already in the results section.*

Our response: This can be easily fixed by being more strict about what should be in the Results section in a revised manuscript.

**References**

Bastian, L., Vigier, N., Reynaud, S., Kerros, M.-E., Revel, M., and Bayon, G.: Lithium Isotope Composition of Marine Biogenic Carbonates and Related Reference Materials, Geostandards and Geoanalytical Research, 42, 403-415, 10.1111/ggr.12218, 2018.

Carignan, J., Vigier, N., and Millot, R.: Three secondary reference materials for lithium isotope measurements: Li7-N, Li6-N and LiCl-N solutions, Geostandards and Geoanalytical Research, 31, 7-12, 2007.

Fritz, P., and Katz, A.: The sodium distribution of dolomite crystals, Chemical Geology, 10, 237-244, https://doi.org/10.1016/0009-2541(72)90005-8, 1972.

Hoffman, P. F., Halverson, G. P., Domack, E. W., Husson, J. M., Higgins, J. A., and Schrag, D. P.: Are basal Ediacaran (635 Ma) post-glacial "cap dolostones" diachronous?, Earth and Planetary Science Letters, 258, 114-131, 2007.

Kunzmann, M., Halverson, G. P., Sossi, P. A., Raub, T. D., Payne, J. L., and Kirby, J.: Zn isotope evidence for immediate resumption of primary productivity after snowball Earth, Geology, 41, 27-30, 10.1130/g33422.1, 2013.

Land, L. S., and Hoops, G. K.: Sodium in carbonate sediments and rocks; a possible index to the salinity of diagenetic solutions, Journal of Sedimentary Research, 43, 614-617, 1973.

Liu, C., Wang, Z., Raub, T. D., Macdonald, F. A., and Evans, D. A. D.: Neoproterozoic cap-dolostone deposition in stratified glacial meltwater plume, Earth and Planetary Science Letters, 404, 22-32, 10.1016/j.epsl.2014.06.039, 2014.

Marriott, C. S., Henderson, G. M., Crompton, R., Staubwasser, M., and Shaw, S.: Effect of mineralogy, salinity, and temperature on Li/Ca and Li isotope composition of calcium carbonate, Chemical Geology, 212, 5–15, 2004.

[Figure]

Pistiner, J. S., and Henderson, G. M.: Lithium-isotope fractionation during continental weathering processes, Earth and Planetary Science Letters, 214, 327-339, 2003.

Rose, C. V., and Maloof, A. C.: Testing models for post-glacial 'cap dolostone' deposition: Nuccaleena Formation, South Australia, Earth and Planetary Science Letters, 296, 165-180, 10.1016/j.epsl.2010.03.031, 2010.

Taylor, H., Kell Duivestein, I., Farkas, J., Dietzel, M., and Dosseto, A.: Lithium isotopes in dolostone as a palaeo-environmental proxy – An experimental approach, Climate of the Past, 2019.
* * *
[Figure]

**Fig. 1.**

[Figure]

[Figure]

**Fig. 2.**

EC-5

SEM image
Mg

EC-9

SEM image
Mg
Ca

**Fig. 3.**